# Direct foliar phosphorus uptake from wildfire ash

Anton Lokshin[1,2], Daniel Palchan[2], Avner Gross[1]

[1] The Department of Environment, Geoinformatics and Urban planning Sciences, Ben Gurion University of the Negev, Israel

[2] The Department of Civil Engineering, Ariel University, Israel

*Correspondence to*: Anton Lokshin (lokshinanton@gmail.com)

**Abstract**. Atmospheric particles originating from combustion byproducts (burned biomass or wildfire ash) are highly enriched in nutrients such as P, K, Ca, Mg, Fe, Mn, Zn, and others. Over long timescales, deposited wildfire ash particles contribute to soil fertility by replenishing soil nutrient reservoirs. However, the immediate nutritional effects of freshly deposited fire ash on plants are mostly unknown. Here we study the influence of fire ash on plant nutrition by applying ash separately on a plant's foliage or onto its substrate around the roots. We conducted experiments on chickpea model plants under ambient and elevated $CO_2$ levels, which reflect current and future climate scenarios. We found that plants can utilize fire ash P and Ni through their leaves, by a direct nutrient uptake from particles captured on their foliage, but not via their roots both under ambient and elevated $CO_2$ levels. These results indicate that over a short timescale, plants effectively uptake P from fire ash only via the foliage rather than the root pathway, possibly due to low bioavailability or limited contact between fire ash particles and the roots. According to many previous studies, elevated levels of $CO_2$ will reduce the ionome of plants, due to the partial inhibition of key root uptake mechanism, thus increasing the significance of foliar nutrient uptake in a future climate. Furthermore, the role of fire ash is expected to increase in the future world, thus giving a competitive advantage to plants that can utilize fire ash P from the foliar pathway, as fire-ash P being a particularly efficient and important source of P.

## 1.     Introduction

Atmospheric particles are a major source of macro and micronutrients such as phosphorus (P), iron (Fe), magnesium (Mg), calcium (Ca), potassium (K), manganese (Mn), zinc (Zn) and nickel (Ni) that are essential for terrestrial and marine ecosystems (Mikhailova et al., 2019; Aciego et al., 2017; Chadwick et al., 1999; Goll et al., 2022; Gross et al., 2015, Palchan et al., 2018). In terrestrial ecosystems, particularly the P-depleted Amazonian rainforest, the aeolian contribution of P is of special importance. The depletion of P in the Amazon is driven by processes such as leaching and plant uptake, coupled with the rapid fixation of P to soil minerals. Globally, P deficiency is prevalent due to insufficient P in the bedrock or slow weathering (Aciego et al., 2017), high leaching, or increased plant uptake (Vitousek et al., 2010; Gross et al., 2016; Lynch, 2011; Cunha et al., 2022; Hou et al., 2020). Combustion byproducts from biomass burning ("fire ash") are among the most dominant atmospheric particles with global amounts of 750 Tg and can be deposited locally or transported remotely and supply nutrients even between continents (Barkley et al., 2019). More than 50% of the fire ash particles are generated via high grassland and savanna fires in Africa whereas the rest are emitted from the savannas in the Amazonian region or various drylands (Bauters et al., 2021; Yadav and Devi, 2019). For example, Barkley et al. (2019) have shown

that African fire ash is wind transported and supplies up to half of the P deposited annually in the Amazon basin. Bauters et al. (2021) showed that biomass burning in Africa is a major contributor of P to nearby equatorial forests by atmospheric deposition. According to their findings, the actual deposition of fire ash is higher over the forest because of canopy complexity of the trees, pointing out the importance of the canopy trapping as a pathway for nutrient input into forest ecosystems. However, while the long-term impact of fire ash P reservoirs is well documented, the direct impact of freshly deposited fire ash on plant nutrition has been so far overlooked. Recent studies showed that atmospheric deposition of desert dust has a direct and immediate impact on plant's nutrition by foliar P uptake from particles that captured on their leaves (Gross et al., 2021; Starr et al., 2023, Palchan et al. in review), even though their P is found mostly in biologically unavailable forms (Gross et al., 2015; Longo et al., 2014; Zhang et al., 2018). These studies suggest that plants facilitate P uptake from dust by enhancing the dissolution of insoluble P bearing minerals via exudation of P solubilizing organic acids and acidification of the leaf surface area.

The nutritional impact of fire ash may be even higher than that of dust and volcanic ash as the concentration of P in fire ash particles can reach up to 10000 ppm, which is 5-10 times larger than in desert dust (Tan & Lagerkvist, 2011; Tiwari et al., 2022; Barkley et al., 2019). The knowledge regarding the bioavailability and chemical speciation of P in fire ash are still poorly resolved, yet, solubility is supposedly significantly higher than that of desert dust and volcanic ash, owing to P releasing chemical reactions that occur in high temperatures during biomass burning such as the high melting crystalline phases (calcium/magnesium potassium phosphates) during the combustion (Tan & Lagerkvist, 2011; Tiwari et al., 2022; Myriokefalitakis et al., 2016; Anderson et al., 2010; Wang et al., 2015).

The contribution of fire ash as a nutrient supplier is expected to further increase under almost all future climate scenarios as increased temperatures and drought frequency intensify fire weather conditions, which in turn, escalate the number of fires and their intensities (Liu et al., 2014, 2010; Pausas and Keeley, 2021; Canadel et al., 2021). Thus, understanding the actual nutritional impact of fire ash on plants becomes even more relevant. Furthermore, elevated levels of $CO_2$ generate a phenomenon known as the "$CO_2$ fertilization effect" which accelerates primary biomass production because of the increase in $CO_2$ assimilation by plants (Loladze, 2002; Myers et al., 2014; Gojon et al., 2022). The larger biomass of plants will increase their demand for P and other nutrients to sustain the stoichiometric imbalance caused by $CO_2$ fertilization, making alternative nutrient sources such as fire ash more important.

A recent study that examined the impact of desert dust and volcanic ash on plant's nutrition emphasized the dominant role of the foliar nutrient uptake pathway under elevated levels of $CO_2$ ($eCO_2$) conditions, which are projected to reduce plant nutrient uptake from the root systems causing a reduction of plant nutrient status (Palchan et al., (in review)). Other studies documented that plants that grow under $eCO_2$ increase their efflux rates of soluble sugars, carboxylates, and organic acids such as oxalate and malate that promote the dissolution of nutrient-bearing minerals (Dong et al., 2021)

Here, we aim to study the immediate (i.e., several weeks) impact of fire ash on plant nutrition under current and future $CO_2$ levels. We used chickpeas as our model plants since these plants have a unique set of properties that enhance the dissolution of nutrients from leaves or roots, such as high secretion of organic acids that lower the pH of their leaf or soil surroundings and a high concentration of trichomes on leaves that help them retain particles on their surfaces (Gross et al., 2021; Starr et al., 2023; Yoshida et al., 1997). The plants were grown under $eCO_2$

and ambient $CO_2$ ($aCO_2$) conditions in a greenhouse. The plants were supplied with fire ash as their sole P source, applied both directly to the foliage and to the substrate around the root system, to assess the P acquisition pathway that plants utilize from fire ash deposition. Additionally, we compare the impact of fire ash particles on plant nutrition to that of other atmospheric particles such as desert dust and volcanic ash. To further understand the impact of $eCO_2$ conditions on plants' ability to uptake nutrients, we grew chickpea plants under $eCO_2$, assessing the influence of fire ash P by applying it to the foliage and the substrate around the root system of the plants.

## 2 Experimental Design

### 2.1 Plant material

The experiments were performed on chickpea plant that served as our model plant in this research (Cicer arietinum cv Zehavit, a commercial Israeli kabuli cultivar). We used chickpea as a model plant because previous works had shown that chickpea positively responds to a foliar application of desert dust and volcanic ash (Gross et al. (2021), & Palchan et al. (in review)). The plants were grown at the Gilat Research Centre in southern Israel (31°210N, 34°420E) in a fully controlled glasshouse with inner division for two rooms 2x4 m, 4 m tall. At both rooms, the temperature was fixed at 25±3°C with relative humidity of 50-60%. The rooms were equipped with computer-controlled $CO_2$ supply system (Emproco Ltd., Ashkelon, Israel) that automatically adjusted the $CO_2$ concentrations in the rooms. One room had $CO_2$ levels of 400 ppm ($aCO_2$) and in the other room $CO_2$ levels of 850 ppm ($eCO_2$) to simulate current and future earth $CO_2$ levels based on high emissions scenario (business as usual, SSP 8.5). Initially, 3 seeds per pot were sown in 54 pots filled with inert soilless media (perlite 206, particle size of 0.075–1.5 mm; Agrekal, HaBonim, Israel). After germination, plants were thinned to one plant per pot. All the pots were supplied with a nutrition solution (fertigation) containing the following elements: nitrogen (N) (50 mg L-1), P (3.5 mg L-1), K (50 mg L-1), Ca (40 mg L-1) Mg (10 mg L-1), Fe (0.8 mg L-1), Mn (0.4 mg L-1), Zn (0.2 mg L-1), boron (B) (0.4 mg L-1), Cu (0.3 mg L-1) and molybdenum (Mo) (0.2 mg L-1). The mineral concentrations were achieved by proportionally dissolving $NH_4NO_3$, $KH_2PO_4$, $KNO_3$, $MgSO_4$ and $NaNO_3$. The micronutrients were supplied in EDTA (ethylenediaminetetraacetic acid) chelates as commercial liquid fertilizer (Koratin, ICL Ltd). The location of each pot within the glasshouse was randomized at the beginning and changed every two weeks over the course of the experiment. The plants were dripped irrigated 4 times per day for 5 minutes, via an automated irrigation system from the germination stage. At 14 day after germination (DAG), when plants were at early vegetative stages (two or three developed leaves), we changed the nutrient media for 42 of the pots to P-deficient media (P concentration of 0.1 mg L$^{-1}$) with similar concentrations for the other elements (-P). Visible P-deficiency symptoms (e.g., chlorosis of mature leaves, slight symptoms of necrotic leaf tips and an overall decrease in biomass accumulation) were observed after 35 DAG. At this stage 12 -P plants were applied with fire ash on the foliage (-P+leaf ash) and 12 into the substrate near the roots (-P+ash to the substrate) while the rest served as control group (-P or +P). Overall, each group had six repetitions, resulting in a total of 54 pots grouped as follows: +P (control group), -P (control group), -P+leaf ash, -P + ash to the substrate.

### 2.2 Fire ash type

The fire ash in this study was produced by burning branches and leaves from five trees: *Hyphaene Thebaica, Olea Europaea, Albizia Lebbeck, Cupressus* and *Pinus*. Samples of *Hyphaene Thebaica, Olea Europaea, Albizia*

*Lebbeck* were collected from the campus of Ben Gurion University (31° 15.680 N, 34° 47.964 E). The fire ash that was added to the plants was produced by burning branches and needles of coniferous trees (Cupressus and Pinus) in a controlled bonfire setting. These branches and needles were collected from trees that grow in Neve-Shalom Forest, Israel (31°821 N, 34°987 E). Later, the ash was burned again in a furnace at 550°c for two hours to achieve complete combustion of the organic material. In the real world, fire ash particles exhibit a wide range of burning completeness and temperatures, along with various types of organic matter, including stems, branches, leaves, fruits, needles, etc. In our attempt to describe and quantify the phenomenon, and recognizing the variability in real-world conditions, we wanted to establish a 'perfect' set of conditions. Complete combustion of organic material results in production and oxidation of volatiles or gases such as $CO_2$, carbon monoxide (CO), methane ($CH_4$) or nitrogen dioxide ($NO_2$), with only mineral residue remaining, i.e., "mineral ash" (Bodí et al., 2014). The ash was processed through a set of sieves to achieve a particle size smaller than 63 μm, size subjected to wind dispersion (Guieu et al., 2010). The chemical and mineralogical properties as well as P concentrations of our fire ash samples, resembles reported values in the literature (Bigio & Angert, 2019; Tan & Lagerkvist, 2011; Tiwari et al., 2022).

## 2.3     Fire ash chemical and mineralogical analysis

To validate the typical chemical, mineralogical, and elemental composition of our fire ash samples and ensure comparability with other studies, we conducted X-ray diffraction (XRD), X-ray fluorescence (XRF) as further explained. Mineralogical analysis of the fire ash was performed with XRD on 2 g of the fire ash using a Panalytical Empyrean Powder Diffractometer equipped with a position-sensitive X'Celerator detector. The data were collected in 2h geometry using Cu Ka radiation (k = 1.54178_A) at 40 kV and 30 mA. Scans were run over c. 15 min over a 2h range between 5° and 65° with an approximate step size of 0.033°. The phase analysis and quantification were performed using MATCH! 2.1.1 powder XRD analysis software with the PDF-2 crystallographic library (release year 2002) and WINPLOTR (September 2018 version) graphic tool for powder diffraction. We based the phase quantification on the relative intensity ratio method, which is determined by maximum intensity peaks relative to the corundum ($Al_2O_3$) maximum intensity peak (Hubbard & Snyder, 1988), using published intensity values for all identified phases. Elemental analysis was performed on 1 g of dried powder using XRF spectrometer (Panalytical Axios X-ray system) with a single goniometer-based measuring channel consisting of a Super Sharp X-ray Tube with a rhodium anode operated at up to 60 kV and up to 50 mA at a maximum power level of 1 kW. The WDXRF used a beryllium (75 mm) anode tube window with different analysing crystals (LiF200 crystal, PE (002) crystal, Ge (111) crystal, PX1 synthetic multilayer monochromator, PX7 synthetic multilayer monochromator) to analyse the range of elements. The instrument is equipped with two types of detectors: a flow detector for longer wavelengths and a scintillation detector for shorter wavelengths. The chemical and mineralogical properties of the fire ash are presented in Table S1 and S2 in the Supplement. The total P and the total elemental analysis in fire ash was measured using ICP-MS (0.625%). We performed a modified Hedley scheme to assess P fractionation in the four fire ash samples following a modified Hedley scheme (Mirabello et al., 2013) (Gross et al., 2016). Briefly, P was sequentially extracted from 0.1g of fire ash in four steps. First, the fire ash P was extracted with 30 ml of 0.5M $NaHCO_3$ ($HCO_3$-P, which reflects the dissolved and labile P, which is considered to have high bioavailability). In the second step, dust P was extracted with 1M NaOH (NaOH-P, considered as P that is loosely sorbed to Fe or Al oxides). In the third step, P was extracted with 1M

HCl (HCl-P, considered as Ca-P complexes). The P in the extracts was measured using the molybdenum blue method (Murphy & Riley, 1962) in a microplate photometer (Multiskan Sky; Thermo Scientific). The chemical and mineralogical properties of the fire ash analogue are presented in the supplement, Tables S1,S4.

### 2.4 Fire ash application

Fire ash was applied manually on the plants in two separate applications either on the foliage or on the roots. The first fire ash application was made after the first visual deficiency signs occurred, 35 DAG. Since there is limited data on the actual deposition of fire ash on terrestrial ecosystems, we followed the application doses of Gross et al. (2021). Briefly, Gross et al mimicked the yearly average amounts of dust deposition (January to April) in western Negev region, Israel (Offer & Goossens, 2001; Uni & Katra, 2017), where the application dose was set

to the equivalent value of 30 g m$^{-2}$, a typical dust deposition level during the major growth period. While this amount is not accurately representing the actual deposition of fire ash, it allows for a comparison of the impact of the two different atmospheric particles. Leaf area was determined in parallel trials in which the chickpea was grown in similar growing conditions. Based on this, 1.5 g of fire ash was applied in each application, between 35 and 45 DAG, resulting in a total of 3 g of ash per plant. The application of fire ash particles was performed in the

following manner: we placed the ash particles in a sieve with a 63-micron mesh size and spread the ash by gently shaking the sieve above each plant, while part of the particles was spilled around the plants during the process. Prior to the application, the pot's surface was covered with nylon to prevent settling of ash particles to the root system. Afterwards, the plants were left undisturbed with the settled ash particles on their foliage. The same amount of ash that was applied to the foliage was applied to the substrate around the root area. Afterword, ash

particles were gently mixed around the roots to enhance the physical contact between the roots and the particles, thereby increasing the chances of having a more significant impact. All the surfaces of all the pots (-P foliar ash, -p root ash, -P and +P) were covered with nylon to equalize plants' conditions and minimize the effects of unrelated processes.

### 2.5 Plant biomass and elemental analysis

The plants were harvested 10 d after the last ash application, which was 55 DAG. Afterwards, we followed a cleaning method used in the study of Gross et al. (2021). Briefly, the plants were rinsed in tap water, 0.1M HCl, and three times in distilled water to remove any remaining ash residues. Then the plants were dried in an oven at 65°C for 72 h and their dry weight was determined. Given that plant growth and alterations in plant ionome are evident through an increase in biomass, our examination focused on the above ground part of the plants (shoot),

including stem, branches and foliage. Sample preparation for mineral analysis was conducted as follows: the dry plants were ground to a fine powder in a stainless-steel ball mill (Retsch MM400; Germany). Elemental analysis was performed by burning 1 g of the plant material in a 550°c for 4 hours, to eliminate the organic material. The ashed plant material was subsequently dissolved using 1ml concentrated $HNO_3$. The solution was diluted with a double distilled water (DDW) to achieve a 1:100 dilution. The obtained solution was then measured by Agilent

8900cx inductive coupled plasma mass spectrometer at the Hebrew University (ICP-MS). Prior to analysis, the ICP-MS was calibrated with a series of multi-element standard solutions (1 pg/ml - 100 ng/ml Merck ME VI) and standards of major metals (300 ng/ml - 3 mg/ml). Internal standard (50 ng/ml Sc and 5 ng/ml Re and Rh) was added to every standard and sample for drift correction. Standard reference solutions (USGS SRS T-207, T-209)

were examined at the beginning and end of the calibration to determine accuracy. The calculated accuracies for the major and trace elements are 3% and 2%, respectively. Biomass and elemental properties of the control plants, root and foliage-treated plants and fire ash particles are given in tables S2-S4 in the supplement.

### 2.6 Leaf pH and Fire Ash Holding Capacity

The pH was measured by manually attaching a portable pH electrode designed for flat surfaces (Eutach pH 150. P17/BNS Epoxy pH electrode with flat head for cream samples and surfaces, refillable) onto the surface of three or four leaves from each plant. pH recordings were taken once a week between the beginning of a P-deficient nutrition until the day before the termination of the experiment (Gross et al. (2021); starr et al. (2023), Yoshida et al., 1997) (Table S5 in the supplement). The measurement of leaf surface pH was made on P-deficient and P-sufficient plants, but not the fire ash treated plants because the presence of the material on the leaf surface interferes with the physical contact between the flat surface of the pH electrode and the leaf itself.

Fire ash holding capacity is a measure for the maximal mass of ash that can be held on the surface of a leaf after application of an excessive dose of ash and removing the free particles. This value was achieved by manually dispersing 1 g of ash on the adaxial surface of 0.5 g of fresh leaves, which were detached at the end of the experiment from 12 specially grown P-deficient and P sufficient plants that did not receive the ash treatment (additional plants). These plants were taken to Ben Gurion biogeochemistry lab to perform the holding capacity analysis .After fire ash application, the leaves were gently shaken for 10 sec and weighed. The differences between the weights before and after the ash application represent leaf holding capacity (Gajbhiye et al., 2016).

### 2.7 Statistical Analysis

Treatment comparisons for all measured parameters were tested using post-hoc Tukey honest significant difference (HSD) tests ($P < 0.05$). The significant differences are denoted using different letters in the figures. The standard errors of the mean in the vertical bars (in the figures) were calculated using GraphPad Prism version 9.0.0.

## 3 Results

### 3.1 Fractionation of P in fire ash samples

Most of the P in fire ash that we produced from four different tree types was found in the HCl fraction (ranging from 59%-80%). The fraction of $NaHCO_3$-P ranged from 18%-39%. The NaOH-P fraction ranges from 1%-2%. The average total P ranges from 5382 mg/g to 7323 mg/g (Fig. 1).

### 3.2 Shoot biomass and total P under $aCO_2$

Plants that received foliar application of fire ash show a significant increase in their biomass and P content (Fig. 2 b,d). Control plants had a dry biomass of 1.36 g, while the foliar treated plants had a dry biomass of 2.14 g, indicating an increase of 57.0%. Similarly, the P content increased from 0.96 mg to 1.45 mg, representing a growth of 50.3%. No increases in biomass or P content were observed in plants that received fire ash in the substrate (Fig. 2a,c).

### 3.3 Shoot biomass and total P under eCO₂

Plants that received foliar application of fire ash show a significant increase in their biomass and P content (Fig. 3b,d). Control plants had a dry biomass of 1.50 g, while the foliar treated plants had a dry biomass of 1.84 g, representing an increase of 23.2%. Additionally, the P content rose from 1.03 mg to 1.17 mg, reflecting an increase of 12.6%. No increases in biomass or P content were observed in plants that received fire ash in the substrate (Fig. 3a,c).

### 3.4 Shoot nutrient status under eCO₂ conditions

To quantify the impact of elevated $eCO_2$ conditions on the nutrient status of plant samples, we conducted a comparison between the ionome of chickpea plants grown under $aCO_2$ and those grown under $eCO_2$ levels for both foliar fire ash treated and untreated plants. The comparison was conducted as follows: the average value of each nutrient in plants grown under $aCO_2$ was calculated, and then each nutrient in individual chickpea plants grown under $eCO_2$ levels was expressed as a ratio relative to the average under $aCO_2$ conditions ($eCO_2$ plant$_{(each\ individual\ plant)}$/$aCO_2$ plant $_{(average\ of\ all\ the\ control\ plants)}$). We observed that $eCO_2$ conditions led to a reduction in the concentrations of Mg, K, Ca, Mn, Zn, Cu, Fe, and Ni, ranging from 28.7% (Fe) to 90% (Ni). The foliar fire ash treated plants indicate a significant increase in the concentration of Ni, offsetting the reduction of the control untreated plants that was seen under $eCO_2$ from -90% to -60%, representing a 33% increase. No significant changes were observed for other nutrients (Fig. 4).

### 3.5 Leaf surface properties

Leaf surface pH ranged from 1 to 1.3 with an average value of 1.18 (Table S5 in the supplement). Fire ash holding capacity ranged from 0.07 g to 0.17 g with an average value of 0.13 g (presented in table S6 in the supplement). Fire ash holding capacity values were at the same order of magnitude as the values of desert dust and fire ash that were reported in Palchan et al. (in review).

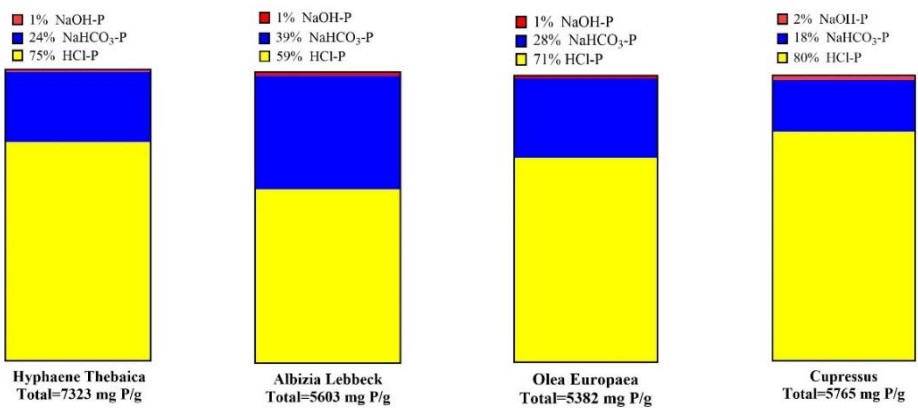

**Figure 1: The fractionation of P in fire ash particles. Yellow color represents P fraction released by 1M HCl, blue color represents P fraction released by 0.5M NaHCO₃ and red color represents P fraction released by 1M NaOH. The average total P value of the four samples is 6018 mg P/g.**

**Root treatment 412 ppm (aCO₂)** **Foliar treatment 412 ppm (aCO₂)**

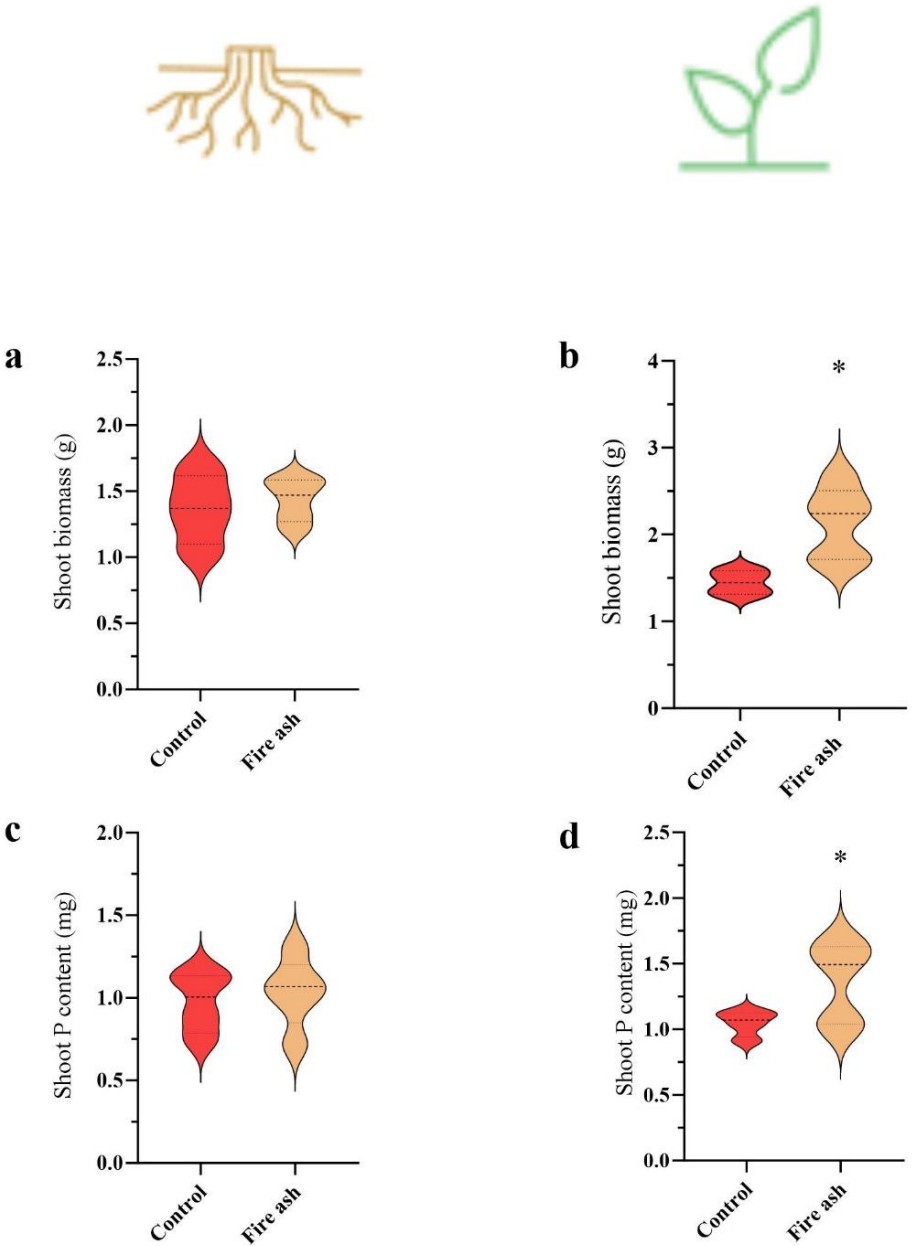

Figure 2 (a-d): The biomass and P content in chickpea plants that were grown under aCO₂ levels, after application of fire ash in the substrate or to the foliage. Control plants with no ash application colored in red while plants treated with fire ash are indicated in a bright brown color. (a,b) Shoot biomass of plants that were applied with fire ash on their roots or their leaves. (c,d) P content of plants that were applied with fire ash on their roots or their leaves. Asterisks represent statistically significant differences between bars (P<0.05, Tukey test). Error bars represent standard deviations (n = 5). The results are presented in a Violin plot, where the dashed line represents the median, and thin dotted lines indicate quartiles.

**Root treatment 850 ppm (eCO$_2$)**   **Foliar treatment 850 ppm (eCO$_2$)**

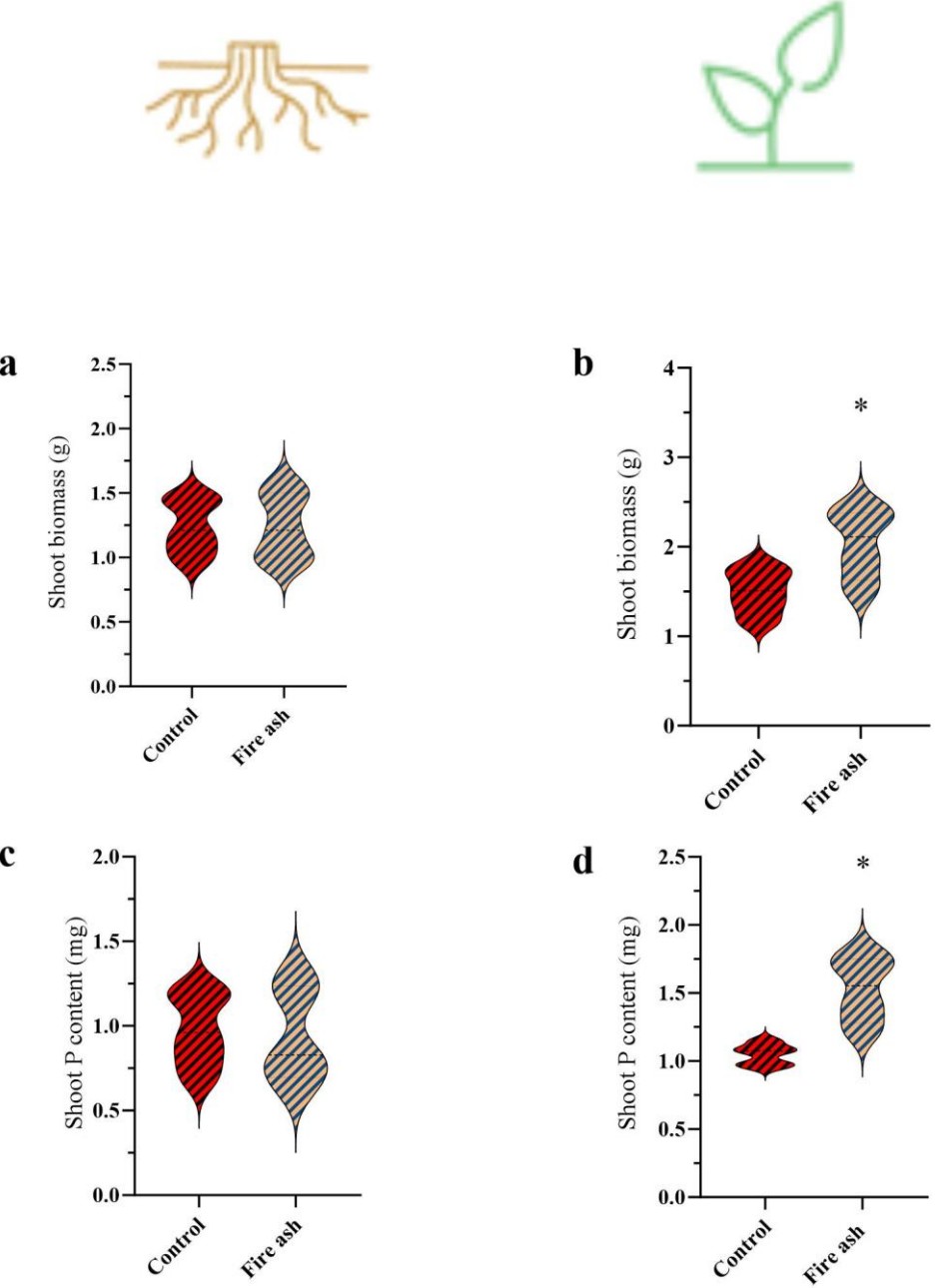

**Figure 3 (a-d): The biomass and P content in chickpea plants that were grown under eCO$_2$ levels, after application of fire ash in the substrate or to the foliage. Control plants with no ash application colored in red, and plants that were applied with fire ash are colored in bright brown. (a,b) Shoot biomass of plants that were applied with fire ash on their roots or their leaves. (c,d) P content of plants that were applied with fire ash on their roots or their leaves. Asterisks represent statistically significant differences between bars (P<0.05, Tukey test), n = 5. The results are presented in a Violin plot, where the dashed line represents the median, and thin dotted lines indicate quartiles.**

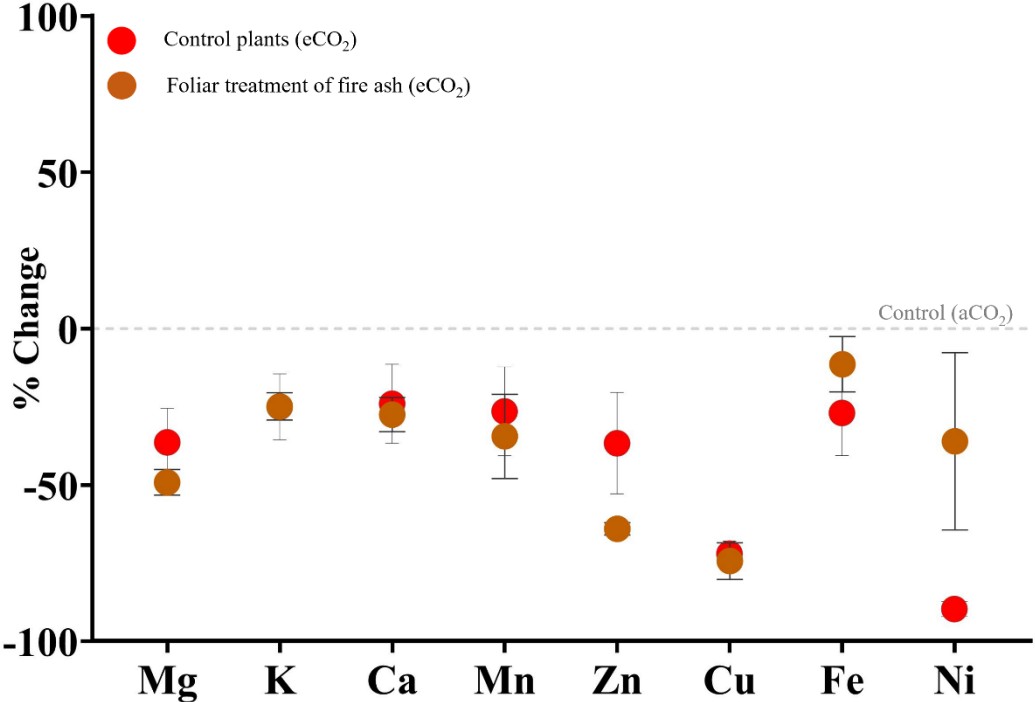

**Figure 4: The % change in the nutrient concentration of plants that were grown under eCO$_2$ conditions in comparison to the plants that were grown under aCO$_2$ conditions. Red circles represent untreated control plants. Brown circles represent plants that received fire ash through foliar application. The dashed line represents the average values of the control plants grown under aCO$_2$ levels; the values below the dashed line indicate a decline in the respective nutrient under eCO$_2$ levels. Error bars represent standard deviations (n = 5).**

## 4    Discussion

### 4.1    XRD, XRF and total P

The XRF, XRD, and total P values of the fire ash samples in our study are within the same order of magnitude as those found in the ash of Mediterranean Savanna forests (Sánchez-García et al., 2023). Also, the fractionation of P in the ashes aligns with fractionation reported in other studies (Wu et al., 2023). The results demonstrate fire ash exhibit a higher concentration of total P in comparison to soils, mineral dust or volcanic ash (Ranatunga et al., 2009; Tiwari et al., 2022; Starr et al., 2023), with a significant percentage being in a soluble form (18%-39%). Yet, the majority of fire ash P is not readily soluble and is found in biologically unavailable form.

### 4.2    P uptake from fire ash under aCO$_2$ levels

Foliar application of fire ash under aCO$_2$ levels increased chickpea biomass and total P content compared to untreated control plants, demonstrating that foliar uptake of P from fire ash has a direct nutritional impact on plants, providing P for biomass growth and boosting photosynthesis. Chickpea plants were more responsive to

fire ash in comparison to experiments with desert dust and volcanic ash from Palchan et al. (in review), demonstrating the importance of fire ash P for plant growth. However, because there was no nutritional impact when fire ash was deposited in the substrate area near the roots, we conclude the nutritional impact occurred exclusively through the foliar pathway, even though the concentration and solubility of P and other nutrients in fire ash are considered higher in comparison to desert dust and volcanic ash (Bigio & Angert, 2019; Gross et al., 2015; Tan & Lagerkvist, 2011; Tiwari et al., 2022). The low bioavailability of fire ash P for root uptake implies that fire ash P is not fully soluble in the rhizosphere and maybe loosely attached to Ca, Fe or Al, that prevents its utilization (Masto et al., 2013; Qian et al., 2009; Tan and Lagerkvist, 2011; Santín et al., 2018). Another reason for the roots impairment in P uptake might be the insufficient physical contact between fire ash particles and the roots, even though the tested plants exhibited an extensive root system, with an average root-to-shoot ratio of 50:50 and the pH of the substrate was around 7, in the range of most alkaline soils. In contrast, plants foliage has a greater surface area that increases the direct physical contact between fire ash particles and plant tissues, thus creating a suitable condition for their partial solubilization. These results suggest that solubility tests examined by chemical extractions do not necessarily reflect actual biological availability and emphasize the importance of fertilization experiments with plants. It is difficult to compare our results with other studies because direct foliar uptake of P from fire ash particles has been so far overlooked, until now. As far as we know, our results represent the first experiment in which fire ash particles were directly added to plants, thus definitively challenging the common perception that P is solely taken up through the roots from the soil, even in the case of atmospheric particle deposition.

### 4.3    Foliar nutrient uptake mechanism

Gross et al. (2021) and Palchan et al. (in review) have studied foliar nutrient uptake mechanisms and pointed out that the presence of trichomes facilitates the adhesion of dust captured on leaf surface and promotes the release of P solubilizing metabolites, such as malic citric and oxalic acids. In addition to the natural secretion of organic acids, the increased secretion may be enhanced as a result of the breakage of trichomes following the application of atmospheric particles on plants foliage. As in previous studies, we also measured a highly acidic leaf surface environment (average pH value of 1.15) and a high fire ash holding capacity (average value of 15%), thus supporting the previous assertions that low pH and high holding capacity may help facilitate P uptake on plant leaves. The combination of leaf surface acidification, secretion of organic acids and additional exudations combined with an increased trichomes density, enhances capture and holding of particles, which in turn facilitates the foliar nutrient uptake in chickpea (chickpea leaves holding capacity of a fire ash particles are presented in Table S6 in the supplement).

### 4.4    Foliar nutrient uptake under $eCO_2$ levels

In accordance with previous studies, growing chickpea plants under $eCO_2$ levels drove a significant reduction in nutrient status in comparison to plants that were grown under $aCO_2$ levels, despite receiving the same fertigation solution. The nutritional decline under $eCO_2$ mostly occurs due to the 'dilution' effect, where accumulation of carbon (C) exceeds that of mineral nutrients, and more importantly, because of a result of a partial inhibition of key root uptake mechanisms (Myers et al., 2014; Loladze, 2002; Gojon et al., 2022). In our experiment the control group was grown under $eCO_2$ conditions and did not exhibit an increase in biomass. This indicates that the

'dilution' effect is not responsible for the nutrient reduction observed in our study and suggests that the primary reason for the nutritional reduction is likely the downregulation of transfer from roots to shoot or root uptake. As a result of the foliar application of fire ash particles, the nutritional reduction under $eCO_2$ was partially offset, leading to an increase in plants' Ni concentrations (Fig. 3). Yet, the nutritional contribution of foliar fire ash under $eCO_2$ levels was less prominent compared to the foliar application of desert dust and volcanic ash, which in addition to Ni also increased the concentration of Fe (Palchan et al. (in review)). We anticipated that the foliage would absorb additional nutrients, such as K, Ca, and Zn, which are present in higher concentrations in fire ash compared to desert dust and volcanic ash (Table S2). The absence of an impact of fire ash on chickpea Fe concentration contrasts with previous laboratory studies that demonstrated substantially greater Fe solubilities in combustion-derived aerosols compared to crystalline minerals (Schroth et al., 2009; Fu et al., 2012). One reason for the low response to fire ash Fe may be related to incomplete combustion which determines the bioavailability of mineral nutrients in fire ash which is mediated by combustion temperature and the presence of oxygen (Tan and Lagerkvist, 2011). In addition, natural forest fires exhibit varying combustion conditions, adding to the complexity of projecting the actual nutrient bioavailability of wildfires (Bodí et al., 2014). Thus, the actual availability of fire ash nutrients such as Fe, P, and others warrants further research, based on laboratory and field fertilization experiments at different combustion conditions.

### 4.5    Broader aspect

We recognize that in other plants, the phenomenon of foliar uptake might be less pronounced (Starr et al., 2023; Gross et al., 2021). in our opinion, selecting a suitable plant was essential to effectively demonstrate it. In a broader context, numerous articles have documented the contribution of wildfire ash to soil fertility through long-term processes, despite the fact that fire ash P contains a significant proportion of labile P (Bauters et al., 2021; Barkley et al., 2019; Bodí et al., 2014). The current perception dictates that fire ash nutrients are not immediately available to plants but rather interact with soil components and bind to minerals in the soil (Tiwari et al., 2022.; Chadwick et al., 1999; Okin et al., 2004). Our results demonstrated that in the time frame of a few weeks, the contribution of fire ash through the roots is negligible. In contrast, fire ash has an immediate nutritional significance to plants via foliar nutrient uptake pathway. It happens due to the foliar secretion of organic acids, such as oxalic and malic, which facilitate the dissolution of fire ash P. Bauters et al. (2021) emphasized that the canopy of the trees in African tropical forest capture P from biomass burning byproducts, which, upon settling on the ground with rain, contributed P to the nutrition of trees as was evident by larger and denser trees growth. We postulate that at least part of that fire ash P was taken up directly by the foliage. In a future world where the quantity and intensity of fires are expected to rise, alongside increasing demand for P due to the $CO_2$ fertilization effect, the contribution of fire ash to natural ecological systems will increase. The fact that the foliar uptake mechanism remains generally unaffected under $eCO_2$ suggests that this pathway may become crucial. Plants exhibiting the foliar nutrient uptake trait are more likely to benefit from fire ash fertilization in a future world. Another important contribution of fire ash P deposition, is its potential to alleviate the ecological stoichiometric imbalance resulting from anthropogenic nitrogen (N) pollution in soils, which is expected to grow in the next decades (Liu et al., 2013) and disrupt plants and the ecosystem's function. Thus, fire ash may play a larger role in a world that shifts from N to P limitation (Du et al., 2020). Currently, biogeochemical models do not incorporate foliar P uptake from fire ash and other atmospheric particles such as desert dust and volcanic ash. Given the substantial contribution of foliar uptake to

plant nutrition from atmospheric deposition, it is imperative to include this process in global carbon and vegetation models. Such models should take into account canopy surface areas, canopy roughness, soil fertility and geographical distribution of atmospheric particles sources and sinks. This inclusion will lead to a more comprehensive description of the global P cycle.

## 5    Conclusions

We have conducted controlled experiments where we have grown chickpea plants under ambient and elevated $CO_2$ conditions. We fertilized the plants separately on foliage and roots with fresh fire ash and harvested the plants after 4 weeks. We then studied the plants elemental composition and reached the following conclusions:

- Freshly deposited fire ash has a direct impact on plant P nutrition only through the foliar nutrient uptake pathway and not via the root system.
- The root acquisition of a freshly deposited fire ash P, as well as other nutrients is limited, despite the high solubility of fire ash P.
- In general, P uptake from fire ash was higher than that of other natural atmospheric particles such as desert dust and volcanic ash.
- Foliar nutrient uptake from fire ash is sustained even under elevated $CO_2$ conditions, implying that the significance of foliar nutrient uptake from fire ash will increase in future climates. This is expected as the ionome of plants is anticipated to decrease, primarily due to the partial downregulation of the root's nutrient uptake mechanism.

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

### Acknowledgments

A.L thanks financial support provided by and KKL climate team doctoral fellowship. We thank Dr. Yigal Erel and Ofir Tirosh for their support in ICP-MS analyses. The project was funded by the Israeli science foundation (ISF), grant number 144/19.

### Author contributions

A.L performed the growing experiments, sample preparations, wrote the article along with input from all authors. D.P and A.G conceived and supervised the entire project.

### Competing interests

The authors declare no competing interests.

### Data availability

Source data for the main text, methods and supplementary information are provided as supplementary information tables.

### Additional Information

Correspondence and requests should be addressed to Anton Lokshin lokshinanton@gmail.com Daniel Palchan danielp@ariel.ac.il or Avner Gross avnergro@bgu.ac.il.