# Peer review of "Direct foliar phosphorus uptake from wildfire ash"

_EGUsphere, 2023_

## Referee Comment (RC1)

Review for Loksin et al. 2023
Direct foliar phosphorus uptake from wildfire ash

This paper is an important contribution to the biogeochemical cycling community. It shows that phosphorus (P) from atmospheric deposition is taken up by plants through foliage, rather than roots. This has implications for how biogeochemical cycles are represented in models. This work has a few issues and sometimes lacks clarity. Overall, these results are very important to the community, and I suggest publication with minor revisions. I would like to review the manuscript again prior to publication.

**Major issues:**
1. I really appreciate the sequential leach done of the wildfire ash, rather than just soluble and total. However, it is unclear how many samples were run to produce Figure S1. If it is only one, then I recommend at least 2 additional samples are run prior to publication to confirm that this sample is representative of P in fire ash. Please show all fire ash sample results in Fig. S1.
2. Please include a summary of the P in fire-ash results as Fig. 1 in the main manuscript. The atmospheric community will be interested in the results and including this figure in the main manuscript will expand the impact of the work.
3. Similarly, a results and discussion section for fire ash needs to be presented in the main text. Please also discuss how these results compare to previously published P contents and solubilities.
4. The discussion (especially section 4.3) needs clarification and expanding.

**Minor issues:**
The manuscript contained a few careless errors that a thorough proof-reading would have caught prior to submission. For example, at one point the text refers to a figure that doesn't exist. I recommend thoroughly proofing the text prior to resubmission.

**Abstract**
Line 17: change particles to ash for clarity.
Line 19: change "that reflect" to "which reflect"
Line 20: This is a little confusing. Please rewrite for clarity – I think there is a way to only use the word "uptake" once in the sentence.
Line 22: add "the" after In a future climate scenario
Line 24: "with fire ash P being the sole nutrient absorbed by the foliage" – This is a very important finding, but it is unclear if it is P only (as opposed to other elements) or if it is fire ash P (as opposed to other aerosol types like dust).
Line 25: I interpret your data as fire-ash P being a particularly efficient and important source of P. If you agree, please add to the last sentence of the abstract to highlight the significance of the results.

**Introduction**

The intro could benefit from providing some context for the importance of fire as a source of P, particularly to tropical soils that are extremely P-deficient. Even despite tropical soils being depleted in P, they are major carbon sinks, so understanding the biomass response to P deposition to these ecosystems is vital to estimating carbon fluxes accurately. I think a first paragraph around these ideas may highlight the importance of this works' findings and broaden readership.

Line 33: P deficiency is particularly prevalent in tropical soils. Is it really prevalent globally?

Line 34: It is my understanding that P is low in soils because it is leached from soils by precipitation or has been used by plants. The sentence currently reads as "P deficiency is prevalent globally due to its low bioavailability" which doesn't make sense. Please revise for clarity.

Line 38: Savanna's should not be capitalized and should just be "savannas"

Line 37-39: The sentence that starts with "About 65%..." makes it sound like all fire ash particles originate from Africa. Please revise for clarity. I'm not sure what the authors are trying to say.

Line 48: Please do not cite a manuscript under review and take out this paper in the rest of your manuscript. It sounds like it may be accepted soon though. Hope that's the case!

Line 55: There is literature showing that fire ash P is more soluble than dust from Barkley et al., 2019 and references therein). Please update this sentence to reflect this literature.

Lin 61: These papers are ok to cite, but papers from 2014 and 2010 are pretty old in fire science – please add more recent references.

Line 68: Please define eCO2 conditions. I also don't understand why the abbreviation e was chosen. Is there a more intuitive abbreviation that could be used? Does e stand for extreme? Define and explain.

Line 69: Please remove the comma.

Line 74: What is eCO2 and aCO2? This abbreviation should be explained. Is it "actual" and "extreme"?

Last paragraph in introduction (line 73):
    Please exclude your hypothesis from this paragraph (sentence beginning on line 76 to end). It's confusing to read this because some of it is opposite of your results. To keep things

clearer, please just say what the question is. For example: "...applied both directly to the foliage and to the roots to assess how plants use P from fire ash deposition"

Line 91: Remove "had"

Line 103: Please add "day" instead of D

Line 108: as should say "ash"

Line 107: Please adjust grammar to say "At this stage, fire ash was applied directly on to the foliage of 12 -P plants..."

Line 111: What is bone-fire burning?

Line 113: Ash is also singular, so please say "Later, the ash was burned again..."

Line 118: move sentences about Tables S1 and S2 to section 2.3 where you discuss the chemical composition methods.

Line 139: This sentence is repeated above. Remove the above one.

**Methods**
Section 2.3:
        It would be helpful to say give a sentence at the beginning of this section describing why each chemical analysis was chosen. For example, say something like "We performed X analysis to quantify total P and a sequential P leach to estimate the different fractions of P." Why was XRD performed? Why was ICP-MS performed? I imagine ICP-MS was done to determine a total P concentration while the sequential leaching was done to determine each P phase. Please state as such.
        What does each step of the sequential leach tell us? Which is most soluble?

Line 149: "two separate pulses" is confusing. I think you can just say twice or two times.

Line 154: I understand following P deposition estimates from Gross et al. 2021 30 g/m2, but is this deposition rate reasonable for fires? Discuss why or why not. Even if it's not, I think it's ok because it's still important to be able to compare your results to another study.

Line 158: Please adjust the grammar. "The same amount of ash that was applied to the foliage was applied to the roots."

Line 163: Change to "remaining ash" instead of "ash remains"

Line 166: Should say "Elemental analysis was performed..." instead of "the elements measurement"

Line 167: Change "get rid of the" to "eliminate"

Line 168: Delete "to achieve a clear solution"

Line 179: Why only the P-deficient plants? Please discuss the reasoning.

Line 187: You can say "additional holding capacity analysis was performed at Ben Gurion University"

Section 2.6: Reference for pH measurement available? Why was leaf pH measured? What does it tell us?

**Results**
3.1:
- What is shoot? Is that the whole plant or the same as the root? Please define and explain why shoot biomass measurements are made for.
- There is no figure 1f or 1e. Please correct so the text refers to the correct figure.
- Figures 1 and 2
  - These figures need to be explained. Please say that they are violin plots. What does the middle dash represent? What do the other dashed lines represent? It's not standard dev because they are not the same on either side of the center dashed line.
  - Please report the significance and what type of significance test was performed.

3.2:
Please make sure the text refers to the correct figures. There is no figure 2f or e.

3.3:
This paragraph is confusing. Please revise for clarity.
Line 208: Replace "Plant's nutrient status" to "the nutrient status of plant samples"
Line 208-209: This is poorly worded and confusing, but a major result.
Figure 3:
- Remove interpretation from Figure 3 caption (second the last sentence)
- The legend on the plot does not match the description of the legend in the caption. Please revise.
- Why was P not measured and provided on Fig. 3?

**Discussion**
Line 278: I think a better and stronger interpretation of your data is that direct foliar application of fire ash is directly beneficial to plants and increases biomass. The word "emphasizing" makes it sound like the results are not novel. Please link the ffact that biomass increase to the plant taking up atmospheric carbon via photosynthesis.

Line 281: Please delete "… confirming out initial hypothesis that fire ash P is more bioavailable to plants" and remove any mention of the hypothesis. The authors could say here "emphasizing the importance of P for plant growth"

Line 281: Please delete "However, despite its projected bioavailability" and replace with something like "because there was no nutritional impact when fire ash was deposited on roots, we conclude the nutritional impact occurred exclusively through foliar uptake"

Line 282: Please delete the sentence that starts with "This discovery." You do not need to discuss your initial hypothesis. You should instead refer to published literature – how are your results similar or dissimilar to previously published studies? Do your results challenge these studies?
Line 286: Imply should be implies

Section 4.2: Connect to your results again. Do your results agree with other results from the Gross lab?
First sentence in 4.2: You do not need to repeat the same Gross et al. 2021 citation in the same sentence.
I think you need a sentence like "our data showing low pH on plant leaves supports previous assertions that low pH may help facilitate P uptake on plant leaves"

Section 4.3:
Delete discussion of your hypothesis (Line 310). Instead discuss why your results are unexpected based on current literature w/ citations.
Line 308: Should contribution be content? I do not understand this sentence.
This section is generally pretty confusing.
        The results presented in Section 3.3 say that the eCO2 conditions reduced the conc of various elements, so the discuss section should discuss why. I feel like the discussion here is missing.

Section 4.4
What is n.d. on line 328?
Line 326: I think the current state thinking is that soluble nutrients like P are more quickly and easily used by the plants after deposition to the soil. Your results are interesting because they contradict that.
Line 328: Delete "in accordance with the common view"
Line 238: Fire also releases N that contributes to N deposition… There is no current N limitation in terrestrial ecosystems because of anthropogenic emissions.

Please discuss how your results inform biogeochemical models. What do the results say about the need for chemical transport model to capture the physics of deposition onto plant leaves? This means that modelers need to have accurate land type model inputs and need to account for surface roughness. Do models currently take deposition onto leaves into account?

SI:
1. Please redefine all abbreviations (except elemental symbols) in SI (e.g., XRF, etc.).
2. Add longer descriptions of each table.
3. How many fire ash samples were analyzed? Figure S1.

---

## Author Comment (AC1)

Dear editor and reviewers,

We are happy to resubmit our paper "Direct foliar phosphorus uptake from wildfire ash" (EGUSPHERE-2023-2617). We were glad to see that the reviewers appreciate the importance of the work we have done. We thank the reviewers for the time and effort that the reviewers invested in reviewing our manuscript. The comments provided were insightful and constructive, contributing to the overall improvement of the paper. We have made a lot of effort to change the entire manuscript based on these comments. We conducted an additional experiment and are presenting new results from the sequential extraction of phosphorus (P) in leaves from four local trees. In addition, we have improved the introduction and hypothesizes, better clarified the methods, and strengthened the discussion section and toned down the conclusions. We are confident that the revised manuscript is now ready for publication in Biogeosciences.

Our responses to the reviewers are provided below in **bold.** For your convenience, following our responses, you will find the revised version of the manuscript with a "track changes" to make it easier for the reviewers to follow the changes we have made in the text.

*Both reviewers commented on our comparison to our second paper of Palchan et al which is in review. We were hoping the paper would be published by now but the war in Israel delayed the publication process. We attached the final version of the Palchan et al paper for the reviewers and editor eyes only. We are assured that Palchan et al paper will be published soon.

**Reviewer 1**

**Major issues:**

1. I really appreciate the sequential leach done of the wildfire ash, rather than just soluble and total. However, it is unclear how many samples were run to produce Figure S1. If it is only one, then I recommend at least 2 additional samples are run prior to publication to confirm that this sample is representative of P in fire ash. Please show all fire ash sample results in Fig. S1.

   **R: We acknowledge the reviewer's comment. In response, we have sampled a new set of plants and conducted four additional ashing experiments to demonstrate that different ashed materials exhibit similar P fractionation. These results have been incorporated into Figure 1 in the main manuscript, as suggested by the reviewer.**

2. Please include a summary of the P in fire-ash results as Fig. 1 in the main manuscript. The atmospheric community will be interested in the results and including this figure in the main manuscript will expand the impact of the work.
   **R: We accept this suggestion. Thus, we added the Hedley sequence results of four ashed plant samples to the main manuscript in Figure 1.**

3. Similarly, a results and discussion section for fire ash needs to be presented in the main text. Please also discuss how these results compare to previously published P contents and solubilities.

   **R: Results and discussion on fire ash P are now presented in the main text. Our discussion includes comparisons to two highly relevant manuscripts, both released in the last few months: Wu et al. 2023 and Garcia et al. 2023 (P12 L354-360).**

4. The discussion (especially section 4.3) needs clarification and expanding.
   **R: The entire discussion was revised, better focused, and expended as the reviewer suggested (changed to section 4.4) (P13-14, L405-432)**

**Minor issues:**

The manuscript contained a few careless errors that a thorough proof-reading would have caught prior to submission. For example, at one point the text refers to a figure that doesn't exist. I recommend thoroughly proofing the text prior to resubmission.

**Abstract**

Line 17: change particles to ash for clarity.

**R: Changed accordingly (P1 L17)**

Line 19: change "that reflect" to "which reflect"

**R: Changed accordingly (P1 L19)**

Line 20: This is a little confusing. Please rewrite for clarity – I think there is a way to only use the word "uptake" once in the sentence.

**R: Changed accordingly (P1 L19)**

Line 22: add "the" after In a future climate scenario

**R: This sentence is changed from its original version to improve clarity (P1 L25)**

Line 24: "with fire ash P being the sole nutrient absorbed by the foliage" – This is a very important finding, but it is unclear if it is P only (as opposed to other elements) or if it is fire ash P (as opposed to other aerosol types like dust).

**R: The foliage exclusively absorbed P from the fire ash particles. We have adjusted the sentence accordingly. (P1 L21-22).**

Line 25: I interpret your data as fire-ash P being a particularly efficient and important source of P. If you agree, please add to the last sentence of the abstract to highlight the significance of the results.

**R: We have incorporated the suggested sentence into the text (P1 L31).**

**Introduction**

The intro could benefit from providing some context for the importance of fire as a source of P, particularly to tropical soils that are extremely P-deficient. Even despite tropical soils being depleted in P, they are major carbon sinks, so understanding the biomass response to P deposition to these ecosystems is vital to estimating carbon fluxes accurately. I think a first paragraph around these ideas may highlight the importance of this works' findings and broaden readership.

**R: This is an important note. The first paragraph has been revised to acknowledge the impact of fire-derived P deposition on biomass in phosphorus-limited tropical soils (P1 L37-38).**

Line 33: P deficiency is particularly prevalent in tropical soils. Is it really prevalent globally?

**R: You are correct; P deficiency is widespread, primarily in tropical soils but also in other regions. Several studies have reported global P deficiency, extending across various ecosystems beyond tropical climates. Examples include Vitousek et al. (2010), Hou et al. (2020), and others. We have modified the first paragraph of the introduction to emphasize this important point (P1-P2 L39-L43).**

Line 34: It is my understanding that P is low in soils because it is leached from soils by precipitation or has been used by plants. The sentence currently reads as "P deficiency is prevalent globally due to its low bioavailability" which doesn't make sense. Please revise for clarity.

**R: P limitation can arise from various factors, including low P concentration in the soil due to insufficient P in the bedrock or slow weathering, high leaching, or increased plant uptake. Additionally, P deficiency may result from its fixation to soil minerals, reducing its biological availability to plants. In tropical soils specifically, P deficiency is attributed to both low total P in the soil due to leaching and biological uptake, as well as high P fixation. We have revised the text to reflect these nuances (P1-P2 L39-L43).**

Line 38: Savanna's should not be capitalized and should just be "savannas"

**R: Corrected accordingly (P2 L47).**

Line 37-39: The sentence that starts with "About 65%..." makes it sound like all fire ash particles originate from Africa. Please revise for clarity. I'm not sure what the authors are trying to say.

**R: According to the literature, Africa is identified as the largest source of fire ash. We have revised the sentence based on this observation (P2 L46).**

Line 48: Please do not cite a manuscript under review and take out this paper in the rest of your manuscript. It sounds like it may be accepted soon though. Hope that's the case!

**R: We acknowledge the challenge of citing a paper that is not yet published. However, the work by Palchan et al. (in review) represents a written paper awaiting publication due to the complex political situation and ongoing conflict in Israel. This paper provides crucial insights that are highly relevant to our results, and it is anticipated to be published soon.**

Line 55: There is literature showing that fire ash P is more soluble than dust from Barkley et al., 2019 and references therein). Please update this sentence to reflect this literature.

**R: The Berkeley et al. (2019) paper is now included in the text. Also, we have added additional references from the literature that highlight the solubility of fire ash P Myriokefalitakis et al., 2016; Anderson et al., 2010; Wang et al., 2015 (P2 L63,69).**

Line 61: These papers are ok to cite, but papers from 2014 and 2010 are pretty old in fire science – please add more recent references.

**R: We have added a couple of newer papers, enhancing the content with the latest research findings (P2 L71).**

Line 68: Please define eCO2 conditions. I also don't understand why the abbreviation e was

chosen. Is there a more intuitive abbreviation that could be used? Does e stand for extreme? Define and explain.

**R: The "e" is referring to elevated. It is described in the text. Corrected accordingly. (P2 L79).**

Line 69: Please remove the comma.
**R: Corrected accordingly (P2 L80).**

Line 74: What is eCO2 and aCO2? This abbreviation should be explained. Is it "actual" and "extreme"?

**R: The "a" refers to ambient and e to elevated. This is now clarified in the text and the sentence was corrected accordingly. (P3 L89).**

Last paragraph in introduction (line 73):
Please exclude your hypothesis from this paragraph (sentence beginning on line 76 to end). It's confusing to read this because some of it is opposite of your results. To keep things clearer, please just say what the question is. For example: "…applied both directly to the foliage and to the roots to assess how plants use P from fire ash deposition".

**We deleted the hypothesis and changed the paragraph accordingly.  (P3 L84-94)**

Line 91: Remove "had"
**R: Corrected accordingly (P3 L113)**

Line 103: Please add "day" instead of
D
**R: Changed accordingly (P4 L125)**

Line 108: as should say "ash"
**R: Corrected accordingly (P4 L130)**

Line 107: Please adjust grammar to say "At this stage, fire ash was applied directly on to the

foliage of 12 -P plants…"

**R: Corrected accordingly (P4 L129)**

Line 111: What is bone-fire burning?

**R: Corrected to a "The fire ash used in this study was produced by burning branches and needles of coniferous trees in a controlled bonfire setting". (P4 L139).**

Line 113: Ash is also singular, so please say "Later, the ash was burned again…"

**R: Corrected accordingly. (P4 L141).**

Line 118: move sentences about Tables S1 and S2 to section 2.3 where you discuss the chemical composition methods.

**R: Changed accordingly. (P4 L150).**

Line 139: This sentence is repeated above. Remove the above one.

**R: The specified sentence has been removed. (P5 L159).**

**Methods**

Section 2.3:

It would be helpful to say give a sentence at the beginning of this section describing why each chemical analysis was chosen. For example, say something like "We performed X analysis to quantify total P and a sequential P leach to estimate the different fractions of P." Why was XRD performed? Why was ICP-MS performed? I imagine ICP-MS was done to determine a total P concentration while the sequential leaching was done to determine each P phase. Please state as such. What does each step of the sequential leach tell us? Which is most soluble?

**R: An explanation sentence for each method has been added as well as the meaning of each step (P4 L156-158, L175).**

Line 149: "two separate pulses" is confusing. I think you can just say twice or two times.

**R: Corrected accordingly. (P5 L186).**

Line 154: I understand following P deposition estimates from Gross et al. 2021 30 g/m2, but is this deposition rate reasonable for fires? Discuss why or why not. Even if it's not, I think it's ok because it's still important to be able to compare your results to another study.

**R: While reliable data on fire ash deposition is limited, it is well-established that average global dust deposition exceeds that of fire ash (based on geographical location). Leveraging the abundance of reliable data on dust deposition, we opted to use dust deposition amounts as our reference for fire ash. This decision facilitates a meaningful comparison of our results with previous studies. It is essential to acknowledge that this may not precisely represent actual fire ash deposition. Therefore, we have included a brief discussion on this point (P5 L192-193).**

Line 158: Please adjust the grammar. "The same amount of ash that was applied to the foliage was applied to the roots."
**R: This sentence is changed from its original version to improve clarity (P6 L204).**

Line 163: Change to "remaining ash" instead of "ash remains"
**R: Corrected accordingly. (P6 L210).**

Line 166: Should say "Elemental analysis was performed…" instead of "the elements measurement".
**R: Corrected accordingly. (P6 L216).**

Line 167: Change "get rid of the" to "eliminate"
**R: Corrected accordingly. (P6 L217).**

Line 168: Delete "to achieve a clear solution".
**R: Corrected accordingly. (P6 L218).**

Line 179: Why only the P-deficient plants? Please discuss the reasoning.

**R: PH measurements were conducted on P-deficient plants, excluding those treated with fire ash. The presence of the material on the leaf surface interferes with the physical contact between the flat surface of the pH electrode and the leaf itself. Additionally, we measured**

**the pH of P-sufficient plants, and the results were similar. This is now explained in the text (P6 L234-237).**

Line 187: You can say "additional holding capacity analysis was performed at Ben Gurion University"

**R: The sentence changed accordingly (P7 L242-L243).**

Section 2.6: Reference for pH measurement available? Why was leaf pH measured? What does it tell us?

**R: Yes, there are a several studies that reported similar acidic pH of chickpea leaves. References now added to the text (P6 L234). These measurements indicate the acidity of the leaf environment which promotes P dissolution in fire ash (see paper of Tiwary et al. (2022) now added to the text). This can provide insights into the foliar nutrient uptake mechanism as was shown in Gross et al. (2021). An explanation was added in the 'Introduction' section (P3 L85-L88).**

**Results**

3.1:

- What is shoot? Is that the whole plant or the same as the root? Please define and explain why shoot biomass measurements are made for.
  **R: The shoot refers to the aboveground part of the plants excluding the roots. An explanation has been added to the manuscript. (Methods, section 2.5, P6 L212-L214).**

- There is no figure 1f or 1e. Please correct so the text refers to the correct figure.
  **R: Corrected accordingly. (P7 L261-L264).**

- Figures 1 and 2
  - These figures need to be explained. Please say that they are violin plots. What does the middle dash represent? What do the other dashed lines represent? It's not standard dev because they are not the same on either side

of the center dashed line.

**R: The dashed lines in the violin plots represent the median and the dotted lines the quartiles. We added this information in the figure legends (P10 L344-L345 and P11 L352-L353).**

Please report the significance and what type of significance test was performed.

**R: The significance is P<0.05, and the test used was the Tukey test. We added this information in the figure legends (P10 L344-L345 and P11 L352-L353).**

3.2: Please make sure the text refers to the correct figures. There is no figure 2f or e.

**R: Corrected accordingly (figures 2 and 3, P10, P11).**

3.3: This paragraph is confusing. Please revise for clarity.

**R: Whole section 3.3 was rewritten and clarified and changed to section 3.4 (P8 L272-283).**

Line 208: Replace "Plant's nutrient status" to "the nutrient status of plant samples".

**R: The sentence is changed from its original version for clarity (P8 L274).**

Line 208-209: This is poorly worded and confusing, but a major result.

**R: We reworked this section, providing detailed clarification of the calculation and explanations for the results (P8 L269-280).**

Figure 3:

Remove interpretation from Figure 3 caption (second the last sentence)

**R: Removed accordingly (changed to figure 4) (P12 L361).**

The legend on the plot does not match the description of the legend in the caption. Please revise.

**R: Corrected accordingly (figure 4).**

Why was P not measured and provided on Fig. 3?

**The plants were P-starved; therefore, the increase in P was assessed by calculating total P values. In this scenario, the P levels in the ionome cannot demonstrate changes in**

**concentration, as there were no variations due to the fact that all available P being directed toward plant growth. This is why figures 1 and 2 represent the total P values.**

**Discussion**

Line 278: I think a better and stronger interpretation of your data is that direct foliar application of fire ash is directly beneficial to plants and increases biomass. The word "emphasizing" makes it sound like the results are not novel. Please link the fact that biomass increase to the plant taking up atmospheric carbon via photosynthesis.

**R: We agree. The word 'emphasizing' has been changed to 'demonstrating.' Since the P-deficient plants were unable to grow, our results demonstrate that the increase in biomass is attributed to additional P directly taken up through the leaves from the fire ash particles. This P is the limiting factor for growth. Even the plants grown under elevated levels of $CO_2$ were unable to grow without sufficient P in their nutrition. The sentence has been modified to: 'Foliar application of fire ash under ambient $CO_2$ levels increased chickpea biomass and total P content compared to untreated control plants, demonstrating that foliar uptake of P from fire ash has a direct nutritional impact on plants, providing P for biomass growth and boosting photosynthesis' (P12 L373-375).**

Line 281: Please delete "… confirming out initial hypothesis that fire ash P is more bioavailable to plants" and remove any mention of the hypothesis. The authors could say here "emphasizing the importance of P for plant growth".
**R: Corrected accordingly. (P12 L379).**

Line 281: Please delete "However, despite its projected bioavailability" and replace with something like "because there was no nutritional impact when fire ash was deposited on roots, we conclude the nutritional impact occurred exclusively through foliar uptake"
**R: Corrected accordingly. (P12 L379-L380).**

Line 282: Please delete the sentence that starts with "This discovery." You do not need to discuss your initial hypothesis. You should instead refer to published literature – how are your results similar or dissimilar to previously published studies? Do your results challenge these studies?
**R: Direct foliar uptake of P from fire ash particles has been overlooked, until now. Our**

**results represent the first instance in which fire ash particles were directly added to plants, thus definitively challenging the common perception that P is solely taken up through the roots from the soil, even in the case of atmospheric particle deposition. We have changed the sentence accordingly (P13 L394-L399).**

Line 286: Imply should be implies.

**R: Corrected accordingly (P12 L386).**

Section 4.2: Connect to your results again. Do your results agree with other results from the Gross lab?

**R: Yes, our results agree with the previous works of Gross et al. 2021 and Starr et al. 2023. We have added a short discussion and connected our results to their findings (P13 L405-408).**

First sentence in 4.2: You do not need to repeat the same Gross et al. 2021 citation in the same sentence.

**R: Additional citation was removed (section 4.2 changed to section 4.3).**

I think you need a sentence like "our data showing low pH on plant leaves supports previous assertions that low pH may help facilitate P uptake on plant leaves".

**R: Additional sentence was added: "As in previous studies, we also measured a highly acidic leaf surface environment (average pH value of 1.15) and a high dust holding capacity (average value of 15%), support previous assertions that low pH and high holding capacity may help facilitate P uptake on plant leaves" (P13 L405-408).**

Section 4.3:

Delete discussion of your hypothesis (Line 310). Instead discuss why your results are unexpected based on current literature w/ citations.

**R: The discussion of our hypothesis was removed, and the sentence was revised based on your comment (P13 L426-L428, L430-L435).**

Line 308: Should contribution be content? I do not understand this sentence.

This section is generally pretty confusing.

**R: This section has been rewritten and additional explanations were added (P13 L416-L428).**

The results presented in Section 3.3 say that the eCO2 conditions reduced the conc of various elements, so the discuss section should discuss why. I feel like the discussion here is missing.

**R: We expanded the discussion regarding the reasons for the nutritional reduction under elevated levels of $CO_2$ (P13 L416-L428).**

Section 4.4

What is n.d. on line 328?

**R: It is a mistake. Corrected accordingly (P14 L453).**

Line 326: I think the current state thinking is that soluble nutrients like P are more quickly and easily used by the plants after deposition to the soil. Your results are interesting because they contradict that.

**R: Your remark is correct; however, it's important to note that most of the P in fire ash particles is not soluble. Additionally, the soluble portions can interact with other minerals and/or microorganisms within the soil. When particles settle onto plant foliage, they may remain undisturbed and be partially dissolved by foliar organic acids such as oxalic, malic, and citric acids. We have clarified this in the text (P14 L457-L459).**

Line 328: Delete "in accordance with the common view"

**R: Changed accordingly. (P14 L455).**

Line 238: Fire also releases N that contributes to N deposition… There is no current N limitation in terrestrial ecosystems because of anthropogenic emissions.

**R: What we aimed to express is that anthropogenic pollution increases the concentration of nitrogen (N) in soils. Consequently, this elevation can lead to an imbalance in plant stoichiometry. The high phosphorus (P) concentration in fire ash could play a partial role in offsetting this anticipated stoichiometric imbalance. The sentence has been revised and clarified in the text (P14 L466-468).**

Please discuss how your results inform biogeochemical models. What do the results say about the need for chemical transport model to capture the physics of deposition onto plant leaves? This means that modelers need to have accurate land type model inputs and need to account for surface roughness. Do models currently take deposition onto leaves into account?

**R: It is an important observation. We have added a paragraph that underscores the significance of updating biogeochemical models (P15 L470-475).**

SI:

1. Please redefine all abbreviations (except elemental symbols) in SI (e.g., XRF, etc.).

   **R: The abbreviations were redefined.**

2. Add longer descriptions of each table.

   **R: Additional descriptions were added.**

3. How many fire ash samples were analyzed? Figure S1.

   **R: We conducted an additional experiment and analyzed four new samples of tree branches and leaves from four different trees to measure their P concentration and fractionation. See our responses to previous comments.**

**Reviewer 2**

This paper is a nice illustration of the potential P-fertilising effect of ash deposited on plant surfaces. It has been known for a long time that P can be taken up by leaves, but this paper quantifies the importance of this P uptake for a crop species in an experimental setting. While the broader conclusions (that our future world will see more ash deposition and that this may make this biogeochemical pathway quantitatively more important than it has been) is true, I feel that the authors 'over-sell' the idea that it is going to be highly significant for plants. The experiment has done several things that maximise the effect of ash: high ash loads, complete burning of ash to remove organic residues, and importantly, choosing a species that is covered in hairs that contain high concentrations of acids, and that (being a legume) is quite responsive to P fertilisation. I would suggest that the conclusions are toned down.

**R: We thank the reviewer for his comments. We have toned down the conclusions throughout the text.**

I agree with (most of) the comments provided by Reviewer 1 and have avoided repeating the same points. Whilst this manuscript does not have many major issues, I believe that the cited manuscript (Palchan et al., in review) should have been made available to the reviewers. Everything indicates that this manuscript is a companion paper based on the same (or parallel experiment). The amount of data presented in this paper is not large. Whilst I have ticked the box 'minor revision', it is possible that a major revision that includes combining Palchan's manuscript to publish it as a single paper may be better advice.

**We provide here the submitted paper of Palchan et al. for the reviewers and editor eyes only. Our paper describes the impact of fire ash on plants and differs from Palchan et al. which tests the impact of mineral particles like desert dust and volcanic ash which vary in their composition, source material and nature from combustion ash. Thus, merging our results with the data presented in Palchan et al. will generate a substantial amount of data, potentially overshadowing the specific effects of combustion ash on plants. We genuinely believe that the ecological importance and impact of combustion ash will interest a large number of readers, justifying a separate discussion and publication. It's important to note that the publication of the Palchan et al. paper has been delayed**

partially due to the war in Israel. We believe that by spring both papers will be published and enable an appropriate comparison between the two papers. Palchan et al. article is expected to be published in the next couple of months. Once Palchan et al. paper will be published, we can ask the editorial committee of 'Biogeosciences' to change the citation status from 'in review' to the actual published paper. We promise that we will update once the paper is published.

L5: I suggest that the species (chickpea) is included in the title.

R: We thank the reviewer for his suggestion, however, we are certain now that the foliar uptake pathway is viable mechanism in other plants as we had published in other papers (Starr et al. (2023) and Gross et al. (2021)), where foliar nutrient uptake from atmospheric particles such as desert dust, was shown to occur in other crop and tree species as well. Thus, foliar nutrition from fire ash is probably applicable to other plant species. We are concerned that if we include the plant's name in the title, it will narrow down the true meaning of our article, which is broader than a single plant. Yet, we acknowledge the reviewer's remark and have added more mentions of the plant specie in the text in several places, and that chickpea is a plant with unique characteristics (P3 L85-L88 , P14 L448-L449).

L18 and L76: ash applied to the roots or rather to the soil surface?

R: The ash was applied to the perlite and gently mixed around the roots to enhance the physical contact between the roots and the particles, thereby increasing the chances of having a more significant impact. This is now explained throughout the text. For example (P1 L17, P3 L90).

L21: You have not demonstrated that plants cannot take up P through their roots. It didn't happen in your experiment, but it is the main pathway under normal circumstances. The way you formulate it here suggests that they don't.

R: You are correct; the primary pathway for nutrient uptake is through the roots. However, our study focuses on the immediate (termed in the paper as "short time scale") impact of fire ash on plants. Our results indicate that in a short timescale, plants may not efficiently uptake P from fire ash via the root pathway, possibly due to low bioavailability or limited contact between fire ash particles and the roots. In contrast, the direct application of fire ash particles

**to the foliage enables extended and direct contact, increasing the likelihood of foliar P uptake. We have revised the paragraph to make it clearer based on your suggestion. (P1 L22-23).**

L25: This is quite far extrapolated from one unique species to all plants. I'd add the word 'potentially' as a minimum and would suggest to tone this down.

**R: The sentence is changed from its original version to town down the conclusions (P1 L21-29).**

L68: "impair" is quite a strong word in "… impair plant's mechanisms of nutrient uptake …". "… reduce plant nutrient uptake …" may be appropriate?

**R: We replaced 'impair plant's mechanisms of nutrient uptake' with 'reduce plant nutrient uptake' (P2 L80).**

L68: explain that eCO2 and aCO2 are elevated and ambient CO2.

**R: We added an explanation of eCO$_2$ and aCO$_2$ (elevated and ambient co2 levels). (P2 L79).**

L77: "Fire ash impacts will be higher than that of other atmospheric sources due to the higher P concentrations and increased solubility in comparison to desert dust and volcanic ash". You cannot test this hypothesis as the experimental design does not include desert dush and volcanic ash treatments. This hypothesis seems to be formulated for the combined experiments of this paper and that of Palchan et al (in review) …

**R: We removed this and other hypotheses in the introduction. We mentioned the comparison between fire ash and mineral dust in the discussion and refer to Palchan et at. And other papers. Please refer to our previous comment regarding Palchan et al. paper.**

L108: fire ash, not fire as.

**R: Corrected accordingly. (P4 L131).**

L102: bonfire I assume?

**R: You are right. Corrected accordingly. (P4 L139).**

L113: further burning at 550 C reduces the ash to mineral-only ash, but is that the type of ash that is dispersed and deposited in the real world, or is that the less-completely burned ash?

**R: In the real world, fire ash particles exhibit a wide range of burning completeness and temperatures, along with various types of organic matter, including stems, branches, leaves, fruits, needles, etc. In our attempt to describe and quantify the phenomenon, and recognizing the variability in real-world conditions, we wanted to establish a 'perfect' set of conditions. However, this is an important comment, and we added an explanation in the text (P4 L142-145).**

L118: dispersion instead of erosion?

**R: Corrected accordingly (P4 L149).**

L155: "…whereas the chickpea grown in similar growing conditions." Change to "… in which the chickpea was grown …"

**R: We replaced the sentences according to your suggestion. (P5 L194).**

L156: 3 g of ash (not as)

**R: Corrected accordingly. (P5 L195).**

L157: "the ash was gently applied manually on the leaves". More detail is needed. Was it evenly spread, how did you avoid spillage? Did you touch the leaf or let the ash fall on it? Touching the leaf of chickpea damages hairs which release strong acids …

**R: The application of fire ash particles was performed in the following manner: we placed the ash particles in a sieve with a 63-micron mesh size and spread the ash by gently shaking the sieve above each plant. Some particles were spilled during the process. Afterwards, the plants were left undisturbed with the settled ash particles on their foliage. The top of the pots was covered to prevent percolation of fire ash particles to the substrate. We added this explanation in the text (P5-P6 L196-L200).**

L159: "applied to the roots". I strongly request different wording, e.g. soil surface, as I'm sure you didn't apply it to the roots in the way you applied it to the leaves. However, you may not like to use 'soil' for perlite. "Substrate" is an alternative option.

**R: We replaced the wording 'applied to the roots' to 'applied to the substrate around the root system". This is now explained throughout the text. For example (P1 L17, P3 L90).**

L162: "The plants were rinsed in tap water, 0.1M HCl and three times in distilled water to remove any ash remains." The HCl concentration seems very high, do you have evidence that this did not damage the plants? Do you also have (microscopic) evidence that all ash was effectively removed?

**R: Before employing this washing method, we tested it extensively in our preliminary experiments. Additionally, this washing method was used in the work of Gross et al. (2021), where they applied desert dust to plants and microscopy was used to ensure that no damage occurred, and no excess of dust particles remained on the leaf surface. We added this explanation in the text (P6 L208-210).**

L175: Not dust but ash.

**R: Corrected accordingly (P6 L226).**

L188: only ash, not dust?

**R: Only fire ash. Corrected accordingly (P7 L243).**

L189: leaf, not leave.

**R: Corrected accordingly (P7 L245).**

Figures 1 and 2: I think that these figures can be effectively combined. I would add Root biomass too. And I would prefer P concentration rather than P content. Differences in P content seem simply due to differences in biomass, but this can be verified by showing P concentration.

**R: We did not detect changes in P concentration because any additional P was directed to biomass growth since the plants were P starved. Thus, the extra P taken up from fire ash particles is reflected by increased P content rather than P concentration.**

**The combination of Figures 1 and 2 would create an 8-panel figure, potentially complicating the understanding of the differences. Figures 1 and 2 depict two distinct growing settings with different $CO_2$ conditions. We believe that presenting them separately is a more representative way of illustrating the results.**

**Root biomass graph has been added to the supplementary file as figure S1 in the complementary.**

L273: "The depletion of the plants nutrient status is caused by the downregulation of the roots system". This is an interpretation that should not be stated in the figure caption. Lower nutrient concentrations at $eCO_2$ are also due to 'dilution' by carbon-based compounds, including higher concentrations of non-structural carbohydrates.

**R: We removed this interpretation from the figure caption. Since the control group, which was grown under elevated $CO_2$ conditions, did not show an increase in biomass, it indicates that the 'dilution' effect is not responsible for the nutrient reduction in this case. We acknowledge that the dilution affect is a major factor in many cases, and we discuss this in the text based on your suggestion. (P1 L23-L25, P13 L416-424).**

L291: These results suggest that solubility tests examined by chemical extractions do not necessarily reflect actual biological availability and emphasize the importance of fertilization experiments with plants." I agree, but the result remains puzzling, and it's a pity that no attempts were made to look into this further. What was the pH in the perlite substrate? Was there evidence of the ash dissolving and being transported into the substrate? Were there roots throughout the substrate? Chickpea roots have the capacity to access P from poorly soluble salts, through the same mechanism that you propose for leaves.

**R: This is an important observation. The tested plants exhibited an extensive root system, with an average root-to-shoot ratio of 50:50 and the pH of the substrate was around 7. Also, our aim was to study the immediate nutritional impact of fire ash particles right after deposition. Thus, the smaller physical contact between the fire ash particles than that of the leaves is an inherent property in field conditions and in our experimental system. Investigating the physiological response of roots following the application of fire ash and other atmospheric particles is an intriguing question that deserves dedicated research. We clarified this the text (P12 L389-L390).**

L296: " … promotes the release of P solubilizing metabolites, such as malic citric and oxalic acids". As far as I know these compounds are contained in gland hairs, and only released when these break, rather than 'exuded'. Breakge may occur naturally, but I'd suggest that you comment on this rather than suggesting that these compounds are continuously exuded.

**R: According to the work of Gross et al. (2021), chickpea plants sequester oxalic, malic, and citric acids from trichomes independently of their breakage. P-deficient chickpea plants increase the granular secretion by activating different strategies, for example increasing the**

**density of trichomes. However, we agree that fire ash application may physically break the glands and increase the exudate concentration. This is now mentioned in the text (P13 L403-L405).**

L321: "Another possible factor could be the elevated pH level of the fire ash particles which may impact the chemical environment of the leaf surface." But your measurements demonstrate a highly acidic environment …

**R: You are correct. The chickpea leaf environment is highly acidic; however, we did not measure leaf surface pH after the application of fire ash particles as the fire ash interferes with the pH measurements. Thus, we deleted this sentence from the text (P14 L443-L446).**

Figure S1: why do the fractions not add up to 100%? Is that the soluble fraction that's missing? Why not include it?

**R: We have added Figure S1 to the manuscript and incorporated four additional samples, as requested by Reviewer 1. The figure was changed based on your and reviewer 1 suggestion.**

[revised manuscript text omitted]

---

## Author Comment (AC3)

**Main Manuscript for**

Foliar nutrient uptake from dust sustains plant nutrition

Daniel Palchan[1*], Anton Lokshin[1,2], Elnatan Golan[3], Ran Erel[3], Sthephen Fox[4], Daniele Andronico[5], and Avner Gross[2]

1. The Department of Civil Engineering, Ariel University; Ariel, Israel.
2. The Department of Environment, Geoinformatics and Urban planning Sciences, Ben Gurion University of the Negev; Beer Sheva, Israel.
3. Institute of Soil, Water and Environmental Sciences, Gilat Research Center, Agricultural Research Organization; Gilat, Israel.
4. Faculty of Chemistry, Weizmann Institute of Science; Rehovot, Israel.
5. Istituto Nazionale di Geofisica e Vulcanologia, Sezione di Catania-Osservatorio Etneo, Rome, Italy.

* Daniel Palchan.

**Email:** danielp@ariel.ac.il

PNAS strongly encourages authors to supply an ORCID identifier for each author. Do not include ORCIDs in the manuscript file; individual authors must link their ORCID account to their PNAS account at www.pnascentral.org. For proper authentication, authors must provide their ORCID at submission and are not permitted to add ORCIDs on proofs.

**Author Contributions:**
Conceptualization: DP, AG, RE

Dust sampling: AL, DA

Methodology: DP, AG, RE

Investigation: AL, EG, SF

Visualization: DP, AL, EG

Funding acquisition: AG, RE, AL

Project administration: DP, AG

Supervision: DP, AG, RE

Writing – original draft: DP, AG, AL, RE

**Competing Interest Statement:** The authors declare no competing interests.

**Classification:** Physical Sciences - Earth, Atmospheric, and Planetary Sciences; Biological Sciences - Plant Biology.

**Keywords:** plant nutrition; Nd isotopes; hidden hunger; foliage;

**This PDF file includes:**

Main Text

Figures 1 to 5

**Abstract**

Soils are the mineral inventories from which plants uptake nutrients to build their ionome. The root system is considered the exclusive pathway by which plants tap into the soils for mineral-nutrients uptake. Here we show that plants can absorb nutrients directly from mineral dust deposits on their leaves via foliar absorption. We combine plant physiology with chemical and isotopic analyses to describe and quantify ion transfer from mineral dust lying on leaf surfaces to plant tissues, the foliar pathway. By applying desert dust and volcanic ash to various chickpea varieties we show that within weeks, treated plants had grown significantly and derived essential nutrients such as P, Fe, and Ni primarily via their foliage. We show how the foliar pathway is facilitated by leaf chemical and physiological properties such as low pH and trichome densities. Using Neodymium (Nd) radiogenic isotopes, we quantified that nutrient transfer via the foliar pathway accounts for over 60% of our plants ionome, thus overshadowing root absorption. Further, we grew plants in elevated atmospheric carbon dioxide concentrations and found that foliar uptake from dust can offset the nutrient deficiencies induced due to impairing of root-based nutrient uptake mechanisms (1), and the carbon dilution effect. The foliar nutrient pathway also allows plants to overcome soil fertility degradation our world currently faces (2). Collectively, our results offer a newly discovered pathway in plant nutrition and suggest a pivotal role of foliar nutrient uptake. This discovery opens new avenues for addressing anticipated crop nutritional deficits and the broader issue of 'hidden hunger' malnutrition.

**Significance Statement**

It is known that plants rely entirely on soils as their mineral nutrient source. Here, we shed light on an untold pathway for plants to uptake nutrients from mineral dust lying on their foliage. Our work combines plant physiology with biological and geochemical analyses, including radiogenic isotopes, to trace the source of nutrients and to quantify significance foliar nutrient uptake. We show that p leaves morphological and chemical properties facilitate the foliar pathway and that this pathway will become even more pronounced at elevated $CO_2$ conditions. Our results show that mineral dust is used as an alternative nutrient source to plants and describe a possible plant mitigation to deal with declining nutrient status under elevated CO2 world.

**Main Text**

**Introduction**

Plants obtain atmospheric carbon (C) through the foliage while most other resources, such as water and nutrients, are taken up from the soil. Hence, it is generally thought that mineral nutrients such as phosphorus (P), potassium (K), iron (Fe), and other macro and micronutrients are acquired predominantly through the plant's roots system (3). Evidence gathered in recent decades demonstrates that the atmosphere is an important source for mineral-nutrients to terrestrial ecosystem via dust deposition (4–9). However, nutrient uptake pathway from dust through the foliage (i.e., direct foliar nutrient uptake) has been overlooked, even though foliar fertilization is used with synthetic soluble fertilizers in agriculture (10, 11). Mineral dust contribution to plant nutrition was never discussed in the context of direct uptake. Dust contribution was documented to outpace the contribution from weathering of host bedrock in montane environments (12), but this may be solely through the soil and root system by long-term pedogenetic processes. Recently, we designed an experiment where desert dust was applied directly on plants foliage and showed that plants uptake notable amounts of P via their leaves (13). In the context of climate change, the foliar pathway may be even more pronounced for plants that will grow under $eCO_2$ conditions because of two documented phenomena: the 'dilution' effect, where accumulation of C exceeds that of mineral nutrients (1), and more importantly, partial inhibition of key root uptake mechanisms (14), together with soil fertility degradation (2, 15), these changes will drive plants to adapt and look for alternative nutrient uptake pathways. The use of the foliar pathway under $eCO_2$ may offset the alarming phenomenon where an increasing production of carbohydrates causes stoichiometric imbalances with macro and micronutrients such as P, Fe, calcium (Ca), magnesium (Mg), K, zinc (Zn), copper (Cu), nickel (Ni) and others that are vital for the floral ecological systems (16) and for their dependent human and livestock nutrition (2, 17, 18).

Here, we grew chickpea plants in varying $CO_2$ conditions, to demonstrate and describe and quantify the foliar nutrient uptake mechanism. The plants were treated with two types of particles on their foliage that represent the most abundant mineral dust particles in the atmosphere: desert dust and erupted volcanic ash (hereafter "dust"), with average annual emissions of 3000 Tgy$^{-1}$ and 300 Tgy$^{-1}$, respectively (19, 20). We elucidate plant traits that facilitate the foliar nutrient uptake from dust, study its impact on plants ionome, and use Nd radiogenic isotopes to quantify the foliar pathway.

**Results & discussion**

  **Foliar mineral-nutrients uptake**

In our experiments, we simulated dust outbreaks by manually applying them on chickpea plants (*Cicer arietinum cv Zehavit*, a commercial Israeli cultivar). The dust was applied separately on plant roots or directly on its foliage (**Fig. 1**), while control plants were not treated with dust. After several weeks, a significant impact of the foliar treatment was already noticeable where shoot biomass and P content in the foliage-treated plants had increased, following dust treatment, compared with the control group. Desert dust application resulted in biomass and P content

increases of 35% and 21%, respectively, and volcanic ash application resulted in 28% and 35% increases, respectively (**Fig. 1 D, F**). Interestingly, the root-treated plants did not show any increases in the biomass or P content, suggesting that over short timescales (i.e., several weeks), foliar uptake is the only nutrient uptake pathway from freshly deposited dust **(Fig. 1C, E)**. This result was then replicated when similar experiment was conducted with plants grown on sandy soil (**Fig. S1**).

[Figure]

**Figure 1.** Biomass and P content increases due to dust application treatments at $aCO_2$ of 412ppm. (A) Image of experiment setting of the root treatment. (B) Image of experiment setting of foliar treatment. **(C)** Shoot biomass of root treated plants. **(D)** Shoot biomass of foliar treated plants. **(E)** Shoot P content of root treated plants. **(F)** Shoot P content of foliar treated plants. The asterisk denotes statistically significant difference from the control. The biomass and P content in the root treated plants do not show increases compared with the control groups. However, the foliar treatment of both desert dust and volcanic ash caused significant increases in the shoot biomass and P content. This implies that plants acquire P from fresh dust deposits on their foliage and not from the root system. Red color represents control plants, orange desert dust treatment and purple volcanic ash treatment.

**Plant strategies for foliar mineral-nutrient uptake**

Most of the P in the dust is incorporated in the mineral lattice of minerals such as apatite (21), which is largely insoluble under the natural rhizosphere pH range (22). Hence, P in dust has low bioavailability for root uptake. On the leaf surface however, chemical, morphological, and microbial modifications may promote nutrient solubility and bioavailability and thus enable uptake through the leaf surface (13, 23). Examining two contrasting chickpea varieties: wild variety CR934, and common domesticated variety "Zehavit", we found a few properties that facilitate foliar P acquisition from dust **(Fig. 2)**. These include structural, morphological, and chemical modifications that are comparable to those reported in the rhizosphere (22). The foliar-uptake-efficient variety "Zehavit" has significantly more acidic leaf surface (pH ~ 1, **Fig. 2b**), and thus promotes both dissolution and mobility of P from the pH sensitive mineral apatite (24), as well as other mineral-nutrients in the dust (13, 23, 25). Additionally, a unique set of metabolites secreted from the leaf surface augments the foliar uptake pathway. These include increased concentrations of oxalate and malate, which are known to release insoluble P in soils through anion exchange reactions (26, 27), and increased levels of sugars such as glucose and sucrose that may promote the activity of nutrient solubilizing microbes on the phyllosphere (28) **(Fig. 2f, fig. S1).** We further found that increased leaf trichome density on both leaf axial and adaxial sides are associated with increased nutrient acquisition **(Fig. 2c, d,e).** These trichomes facilitate the release of metabolites and promote adhesion of dust captured on leaf surfaces (**fig. S2**) (13). We postulate that other plant species share comparable leaf traits that enhance dust capture and solubility such as wheat and various tree species that showed strong responses to foliar dust fertilization (13, 29). Overall, our results suggest that the combination of leaf surface acidification, secretion of organic acids and additional exudations combined with an increased trichome density enhances foliar dust capture and nutrient uptake in chickpeas.

[Figure]

**Figure 2.** Comparison of two chickpea varieties - CR934 (dotted, pink) and Zehavit (yellow) and their leaf properties under dust foliar fertilization. **(A)** Biomass and P uptake response to foliar dust P. Each column indicates the difference Δ (%) between the foliar dusted plants and the control untreated plants (n=6). **(B)** (I) Leaf surface pH. Each value indicates an average of five measurements on a plant throughout the growth season in control treatment (n=90), and two measurements in foliar dust treatment (n=10). One asterisk indicates significant differences between treatments using a T-test, and a one-way ANOVA (P≤0.05). Three asterisks indicate significant differences between treatments using a T-test, and a one-way ANOVA (P≤0.001). **(C)** Leaf Glandular (black column) and non-glandular (please add a dash to the Non glandular in the figure and change it to Non-glandular) (white column) trichrome density in CR934 and Zehavit control plants (-P and +P). Different letters indicate significant differences between varieties and treatments using Tukey-HSD test (P≤0.05) (n=12). Capital letters refer to non-glandular trichomes and small letters refer to glandular trichomes. **(D)**. SEM scans of non-glandular (red circles) and glandular (yellow circles) trichomes of typical Zehavit leaf. **(E)**. SEM scans of leaves of CR934 (left) and Zehavit (right) varieties. The Zehavit clearly shows higher density of trichomes in the abaxial surface, rendering it as more fit to extract nutrients from dust particles. **(F)**. Exudates of organic acids. Each column indicates the average of leaf washing from four plants, in -P control treatment (n=4). Two asterisks indicate significant differences between treatments using a T-test, and a one-way ANOVA (P≤0.01). Values are concentrations compared with an internal standard.

**Quantifying the contribution of foliar nutrient uptake from dust**

Traditionally, radiogenic Nd isotopes serve as excellent tracers for sources of magmatic rocks (30), sediment archives (8, 9), and water bodies (31). Since Nd is found in high concentration in nutrient bearing minerals (8, 12, 32), Nd isotopes were recently used to trace nutrient sources in plant tissues, where it was shown that the contribution of dust outpaces the weathering of the local bedrock over geological time scales (12, 32). Here, we utilized the ratio of $^{143}Nd/^{144}Nd$ in the εNd notation to trace the source of Nd in our experiments and quantify the flux of dust-borne nutrients such as P or Fe **(Fig. 3)**. We used a two-component mixing model, where the average εNd value of the control plants, -0.3, which arise from the Nd "inheritance" (i.e., the Nd composition of the seed) is regarded as one end member, and dust εNd values are regarded as the second end member, with values of -11 (desert dust) and 5 (volcanic ash). We found that desert dust treated plants were characterized with εNd values of -8.8 to -5, significantly different than the inheritance value of the control group. Similarly, the volcanic ash treated plants were characterized with εNd values of 3.4 to 4, significantly different than the inheritance value of -0.3. Thus, it is evident that the εNd of the foliage-treated plants comprise a mixture of the inheritance and the type of dust applied. Based on the mixing model, the chickpea plant acquired over 60% of its Nd from desert dust deposited on the foliage. Volcanic ash deposited on the foliage contributed over 70% of its Nd (**Fig. 3**). These results imply that Nd isotopes can be used in future studies to quantify the immediate contribution of freshly deposited dust on the plant's nutrition in field and lab experimental settings.

[Figure]

**Figure 3.** Quantification of dust mineral-nutrient flux from the foliage. Radiogenic isotopic ratios of $^{143}Nd/^{144}Nd$ in the different sample groups (x-axis) expressed in $\varepsilon Nd$ values. Diamonds represent the two applied mineral fractions of volcanic ash and desert dust; circles represent plants treated with the dusts and the control groups. Large circles represent plants growing in the 850 ppm $eCO_2$ and small circles represent the 412 ppm $aCO_2$. The color scale reflects the % contribution of Nd originating from the dusts via the foliage, which was calculated using a two-component mixing model. The control plants' Nd signature reflects the inheritance value from the seed, where a value of $\varepsilon Nd$=-0.3 is set as the control, $\varepsilon Nd$=-10.3 as the desert dust value, and $\varepsilon Nd$=4.6 as the volcanic ash value. A foliar contribution of more than 60% is evident in the plants applied with desert dust and more than 70% in the plants applied with volcanic ash. Standard errors on the isotopic values are all smaller than the depicted data points.

**Mitigating eCO₂ induced malnutrition**

We repeated the foliar nutrient uptake experiment in $eCO_2$ conditions of 850 ppm $CO_2$ to mimic high emission scenarios (33) and to examine the foliar pathway under future conditions. Firstly, in accordance with the experiments under $aCO_2$ only foliar-treated plants had significant increases in the biomass and P content while the root treatment influence was negligible. Root treatments

did not show any increases in biomass or P content (**Fig. 4**). The Nd isotopes study had shown that even under $eCO_2$, foliar nutrient uptake efficiency remains prominent **(Fig. 3)**.

**Root treatment 850 ppm**      **Foliar treatment 850 ppm**

[Figure]

**Figure 4.** Biomass and P content increases due to dust application treatments at $eCO_2$ of 850ppm. **(A)** Shoot biomass of root treated plants. **(B)** shoot biomass of foliar treated plants. **(C)** Shoot P content of root treated plants. **(D)** Shoot P content of foliar treated plants. The asterisk denotes statistically significant difference from the control. The biomass and P content in the root treated plants do not show increases compared with the control groups. However, the foliar treatment of both desert dust and volcanic ash caused significant increases in the shoot biomass and P content. This implies that plants acquire P from fresh dust deposits on their foliage and not from the root system. Colors: red – control plants, orange – desert dust treatment, and purple – volcanic ash treatment.

Following this, and in accordance with previous knowledge (18), we found that $eCO_2$ drastically reduced mineral nutrients concentrations of Mg, K, Ca, Mn, Zn and Fe, with devastating reductions in Cu and Ni, by 72% and 90%, respectively (**Fig. 5**). This reduction in concentrations can't be attributed to the dilution effect (18, 34) as biomass in the $aCO_2$ and the $eCO_2$ control groups resemble **(Figs. 1D & 4B)**. In the absence of a significant dilution effect the observed nutrient reduction, under $eCO_2$ conditions, is linked to impairments of mineral nutrient uptake via the root system (14). Surprisingly, the foliar dust-treated plants replenished their Fe and Ni concentrations (both essential micronutrients for plant growth and in human diet) compared with the control group that showed drastic reductions in their ionome. Desert dust treated plants showed increases of Fe and Ni concentrations of 44% and 46%, respectively (**Fig. 5a**). Volcanic ash treated plants showed Fe elevated concentrations of 66% (**Fig. 5b**). The Ni concentrations had more moderate increases from volcanic ash, with 40% higher than in the $aCO_2$. These results stress how much these dust particles are enriched with P and other essential mineral nutrients relative to most soils (7, 35–37), emphasizing the role of the atmosphere as an important mineral-nutrients source, especially to plants growing on soils with low fertility or in dusty regions (12, 35, 36, 38, 39). We surmise that the foliar pathway significance will grow with rising $CO_2$ levels in the atmosphere because of the projected downregulation of the root's nutrient uptake pathway (40). However, the Nd isotopes study does not show this excepted trend and more investigation is required in this aspect.

[Figure]

[Figure]

**Figure 5.** Comparison of the % change in plant ionome of our experiments under various conditions compared with ambient CO2 control plants. Changes in nutrient concentrations of control eCO$_2$ plants (red circles) show that eCO$_2$ conditions deteriorate plant root uptake significantly. **(A)** The effect of foliar treatment of desert dust (orange triangles). **(B)** The effect of foliar treatment of volcanic ash (purple squares). Error bars denote SD. Both desert dust and volcanic ash treated plants show that the ability of plants to uptake nutrients from the foliage will replenish, and even increase, concentrations of Fe and Ni in a possible future earth, hence mitigating the plant nutrient reduction caused by elevated CO$_2$.

**Discussion**

We showed here that dust nutrient uptake via the foliar pathway in chickpea plants plays a major role in their nutrition. Plant foliage captures and dissolves freshly deposited dust particles, making atmospheric mineral nutrients more accessible through the foliage than via the roots. Thus, our findings highlight that dust serves as an alternative source of nutrients to plants on short timescales of few weeks. Furthermore, that foliar dust acquisition compensates for the reduction in nutrients such as Fe and Ni, induced by downregulation of the roots under $eCO_2$ conditions (34).

The broader aspect of our findings emphasizes the central role of the foliar pathway to plant nutrition and nutrient cycles in natural ecosystems, but also relates to livestock and human health, nutrition, and wellbeing. The consequence of Fe malnutrition in humans is anemia that can cause morbidity and even mortality if not addressed (41, 42). Iron deficiency anemia (IDA) affects approximately 15% of the world population with higher risk for children under 5 and childbearing women (42–45). As for Ni, although its biological function is not yet fully resolved, it is mainly found in highest concentrations in nucleic acids, particularly RNA (46). Nutrient deficiency will amplify in a future world where $eCO_2$ will increase the IDA risk factor and Ni deficiency (34, 42). In addition, low plant nutritional values under $eCO_2$ goes beyond agroecosystems and drives stoichiometric imbalances that impact entire food webs, nutrient cycles, and carbon sinks (47–51). Our findings imply that the foliar nutrient uptake pathway from natural dust will play a central role in eCO2 earth, and that this pathway may be a target for novel fertilization techniques to compensate for the expected decline in the crops' nutritional value. The foliar nutrient uptake pathway (foliar fertilization), therefore, may be one essential mitigation mechanism to cope with expected increasing IDA rates and decreasing Ni availability, as well as other nutritional deficiencies.

**Conclusions**

Our experiments in ambient and elevated CO2 conditions, where we had sprinkled dust directly onto plant foliage had resulted in biomass and P content increases compared with our control plants. The physiological and bio-geochemical analyses we conducted on these plants rise the following conclusions:

- The physiological study showed plants secrete low molecular organic acids to lower the leaf surface pH and enable ion mobility. These are exuded from trichomes in higher density that also act as dust traps.
- The geochemical analyses we conducted, characterizing plants ionome of different sample groups, showed which elements are transferred by the foliage.
- The Nd radiogenic isotopic composition had allowed us further to quantify the foliar pathway significance. We found that under the experiments conditions the foliar pathway replenishes over 60% of the plant ionome.
- These results illuminate how plants can uptake mineral nutrients in significant quantities via their foliage, and that chickpea plants, that uptake high concentrations of Fe via this pathway, may help to mitigate malnutrition and IDA.

**Materials and Methods**

**1.1 Plant material and growth conditions**

Two Chickpea varieties (*Cicer arietinum cv Zehavit*) was chosen as our model plant based on their popularity and positive response to foliar dust application (13). The plants were grown at the Gilat Research Center in southern Israel (31°210N, 34°420E) in two separate glasshouse rooms under 13 hours of natural day light and with a fixed temperature of 25±3°C and relative humidity range of 50-60%. The glasshouse rooms were equipped with a computer-controlled $CO_2$ supply system (Emproco Ltd., Ashkelon, Israel) that automatically adjusted the $CO_2$ concentrations. In the first room, $CO_2$ concentration were set to 412 ppm (a$CO_2$) and in the second room to 850 ppm (e$CO_2$), simulating current and future earth $CO_2$ levels based on high emissions scenario (business as usual, SSP 8.5)(52). Following germination, plants were cultivated in 72 pots containing inert media (perlite 206, particle size of 0.075–1.5 mm; Agrekal, HaBonim, Israel).

The pots were randomly placed inside each room and their locations within rooms were changed once a week. Rooms were twice switched over the course of the experiment to avoid bias due to localized conditions.

Plants were fertigated with: N (50 mg $L^{-1}$), P (3.5 mg $L^{-1}$), K (50 mg $L^{-1}$), Ca (40 mg $L^{-1}$), and Mg (10 mg $L^{-1}$). Micronutrients were supplied with EDTA (Ethylene diamine tetra-acetic acid, Koratin, ICL Ltd) for the at concentrations of: Fe (0.8 mg $L^{-1}$), Mn (0.4 mg $L^{-1}$), Zn (0.2 mg $L^{-1}$), B (0.4 mg $L^{-1}$), Cu (0.3 mg $L^{-1}$) and Mo (0.2 mg $L^{-1}$). The plants were drip irrigated three times per day via an automated irrigation system from the germination stage. At 14 days after germination (DAG), when plants were early in the vegetative phase (two or three developed leaves), we changed the nutrient solution of 60 pots to P deficient (P concentration of 0.1 mg $L^{-1}$) to create P starvation (-P treatment) (13). Preliminary tests showed that our -P deficient media allows chickpea plants to continue their growth cycle without gaining additional biomass under e$CO_2$, ruling out the $CO_2$ fertilization effect, which further accounts for the nutrient dilution effect that is common under e$CO_2$ conditions (1, 34). The remaining 12 pots received the same full nutrient media (+P treatment). Plants fertigated with -P solution started to show P-deficiency symptoms such as chlorosis of mature leaves, slight symptoms of necrotic leaf tips and an overall decrease in biomass accumulation at 35 DAG (13). At this stage we applied desert dust and volcanic ash on the -P plants (see section 1.2).

A total of 72 plants were examined in our experiment with 48 treated plants (see section 1.3) and 24 control plants. The treatment refers to desert dust and volcanic ash applied on the plants. Twenty-four plants were applied with dust on their foliage and 24 plants received root treatment followed by gentle mixing of the surface to sink the dust particles deeper. The control plants were 12 plants of +P solution and 12 plants of -P solution. Each treatment group was divided to the two $CO_2$ levels, 36 plants in each. The plants were harvested 10 days after the last dust application (55 DAG). To ensure that nutrients from dust particles applied on roots were not washed by the irrigation during the experiment, we monitored the total P (i.e., P that dissolves in strong acid (7, 53, 54)) in the drainage throughout the experiment.

In a parallel experiment in the same growing rooms, we grew nine additional chickpea plants in a pot containing local, low P sandy soil. This was to test whether natural soil

conditions have any impact on dust nutrient utilization from the roots. This small experiment included three pots of -P treatment, three pots of -P treatment + dust on foliage, and three treatments of -P + dust applied on roots (**Fig. S1**). It is evident that the root treated plants did not grow larger than the foliage treated plants, suggesting that over short timescales, the plants acquire nutrients from freshly deposited dust quicker from the foliage than from the roots (**Fig. S1**).

**1.2 Chickpea lines experiment**

Two chickpea genotypes (*Cicer*) from the Hebrew University of Jerusalem chickpea collection were selected based on preliminary experiments showing contrasting response to eolian dust application to the shoot. The non-responsive genotype: CR205, of the wild progenitor *C. reticulatum* accession, sampled near Savur, Turkey. The responsive genotype 'Zehavit' is a modern, high yield line, and considered popular among the Israeli growers.

From September to November 2022, the experiment was conducted in a sealed climate controlled growing room with natural light at the Gilat Research Center. The temperature was set to 19C° during the day and 16C° at night. The experimental setup included two P levels: 3 mg l$^{-1}$ P (P+) and with low P (P- ~0.2 mg l$^{-1}$). Plants from P- treatment were divided into two groups; the first group was dusted with rock phosphate during the season, and the other group served as a control and did not receive any additional treatment. The three treatments were: P+, DUST P, and P-. One plant from each variety was planted in a 2.3L pot, containing perlite substrate. Each genotype treatment had six repetitions that adds up to 72 pots.

Two weeks after germination, when the P- plants showed P deficiency symptoms expressed in reduce biomass and yellowing of the mature leaves, we dusted the DUST P plants with 1g of rock phosphate (14.07 ± 0.12 total %P), and the second time was a week afterward. The substrate was covered with paper to ensure that the DUST P would not leach to the root system. One day after dust foliar application, we evaluated the dust holding capacity (SI-2).

**1.3 Dust types and its application**

The plants were applied with desert dust and volcanic ash, the two main mineralogical dust types in the atmosphere (55). To achieve enough mass for our experiment, we produced dust analogs [11] from surface desert soil and surface volcanic ash soil, following common procedures described by others (13, 56). The desert dust analog surface soil was collected from the southern Israel Negev desert (30°320N 34°550E) (13). Chemical and mineralogical properties of the resulted dust are comparable to dust collected in the Sahara and other places in the Middle East (9, 57). The volcanic ash analog was collected from Mount Etna (Sicily, Italy) two month after the eruption of February 2022. The ash was taken from the upper cable car station "Finuvia dell'Etna" (37°704N, 14°999E). The samples were then processed through a setup of sieves to achieve a particle size smaller than 63$\mu m$ that are considered windblown (58). The chemical and mineralogical properties of the dust analogs are presented in SI Table 1.

To mimic dust deposition which typically occurs during a few major desert storms or volcanic eruption each year, we applied the dust in two equivalent doses between 35-42 DAG. Total application mass was 3g per plant, to simulate the total dust deposition per m$^2$ for an average

growth period in southern Israel (13). Dust treatments were done either directly on the foliage while covering the pot, preventing the dust to touch the roots, or directly on the roots where the pots were subsequently covered with nylon to equalize conditions with the foliage treated plants. Dust treatment was done by manually sprinkling the dust through a $63\mu m$ sieve in proximity to the foliage.

**2. Analyses**

**2.1 Plant biomass and elemental analysis**

After harvesting, the plants were separated for roots and shoots, washed in 0.1M HCl and rinsed three times in distilled water to remove dust particle residue (13). Later, plant tissue was dried, weighed, ground to powder and dry ashed at 550 C° in a furnace for four hours (26). Approximately 1g of the ashed material was subsequently dissolved using 1ml concentrated $HNO_3$ to achieve a clear solution. To prepare the dust types for elemental analysis, the samples were dissolved on a hotplate by sequential dissolution using concentrated $HNO_3$, HF, and HCl, resulting in clear solutions (9). The elemental composition of the plants, dusts and nutrient solution were analyzed at the Hebrew University using a ICP-MS (Agilent 8900cx; Agilent Technology). Prior to analysis, the ICP-MS was calibrated with a series of multi-element standard solutions (1 pg/ml - 100 ng/ml Merck ME VI) and standards of major metals (300 ng/ml - 3 mg/ml). Internal standard (50 ng/ml Sc and 5 ng/ml Re and Rh) was added to every standard and sample for drift correction. Standard reference solutions (USGS SRS T-207, T-209) were examined at the beginning and end of the calibration to determine accuracy. The calculated accuracies for the major and trace elements are 3% and 2%, respectively. Biomass and elemental properties of the dust analogs, control plants, and dust root and foliage-treated plants are given in SI Tables 1-3.

**2.2 Leaf surface pH**

Leaf surface pH was measured by manually attaching a portable pH electrode designed for flat surfaces (HI-1413; HANNA pH instruments) onto the surface of three leaves from each plant. The measurements were performed four times throughout the growing season (19, 24, 35 and 40 DAG) in the morning, two hours after sunrise.

**2.3 Trichome density**

Trichome density was determined in four young, fully developed leaves from four different plants per variety in the P- treatment only (n=16). Leaves were scanned in a scanning electron microscope (VEGA3; Tescan, Czech Republic). From each leaf, three photos of a $1mm^2$ field were taken, and glandular and regular trichomes were counted.

**2.4 Leaf exudates**

For analysis of the organic exudates, 2g of fresh leaves were sampled randomly from the P+ and P- treatments before harvesting. The leaves were rinsed in 2 ml of distilled water and methanol (50:50) for 10 s. The extracted surface metabolites were supplied with 50 µl of internal standard (ribitol, 0.2 mg ml$^{-1}$) and stored at -80°C until analysis. Before analysis, the extracted samples were vacuum dried overnight at 35°C. The dried material was redissolved in 40 µl of 20 mg ml$^{-1}$ methoxamine hydrochloride ($CH_3ONH_2$ HCl) in pyridine ($C_5H_5N$) and derivatized for 90 min at 37°C, followed by a spike of 70 µl MSTFA (*N*-methyl-*N* (trimethylsilyl) trifluoroacetamide ($CF_3CON(CH_3)Si(CH_3)_3$) at 37°C for 30 min. The dissolved metabolites were then introduced to a mass spectrometry gas chromatograph (Agilent 6850 GC/5795C; Agilent Technology) for analysis. The metabolites were detected by a mass spectrometer, where 1 µl of each sample was injected

in split-less mode at 230°C to a helium carrier gas at a flow rate of 0.6 ml min$^{-1}$. GC processing was carried out using an HP-5MS capillary column (30 m 9 0.250 mm 9 0.25 μm) and the spectrum was scanned for *m/z* 50–550 at 2.4 Hz. The ion chromatograms and mass spectra obtained were evaluated using the MSD CHEMSTATION (E.02.00.493) software, and sugars and amino acids were identified via comparison of retention times and mass spectra with certified GC plant metabolite standards (Sigma Aldrich).

**2.5 Nd isotope chromatography and analysis**

Nd isotopes were measured on the dust and on the plant material. Nd was separated from the samples using TRU followed by LN-spec resins (59). Measurements of the isotopic ratios were performed using a Thermo Neptune multi-collector ICP-mass spectrometer at the Weizmann Institute of Science. A JNdi Nd standard bracketed the samples, resulting with $^{143}$Nd/$^{144}$Nd value of 0.512035 ± 1$^{-5}$ (2$\sigma$, n=60). The data was normalized to $^{143}$Nd/$^{144}$Nd = 0.512115 (60). Rock standards samples of BCR-2 were dissolved and measured along with the plant and dust samples yielding $^{143}$Nd/$^{144}$Nd value of 0.512628 ± 6 (2$\sigma$) that agrees with $^{143}$Nd/$^{144}$Nd = 0.512637 ± 13 value of BCR-2 (61) . The Nd isotopic ratio is expressed as:

$$\varepsilon Nd = \left( \frac{\left( {}^{143}Nd \big/ {}_{144}Nd \right)_{Sample}}{\left( {}^{143}Nd \big/ {}_{144}Nd \right)_{CHUR}} - 1 \right) * 10{,}000$$

where the present value of $^{143}$Nd/$^{144}$Nd = 0.512638 in CHUR (62). A sample isotopic characterization is given in SI Table 4.

Foliar contribution percentages were calculated using simple mixing equation of two components:

$$\% \, Foliar \; contribution = \frac{\varepsilon Nd_{sample} - \varepsilon Nd_{control}}{\varepsilon Nd_{end \, member} - \varepsilon Nd_{control}} * 100$$

Where $\varepsilon Nd$ end member values of desert dust and volcanic ash are -10.3 and 4.5, respectively (Table SI-4 & Fig. 4). Due to the large variation in the control plant $\varepsilon Nd$ values, we decided to present a conservative calculation, where the values of the control were $\varepsilon Nd$ = -1.7 when computing the mixing with desert dust, and $\varepsilon Nd$ =1.3 when computing mixing with volcanic ash.

**2.6 Mineralogical analysis**

Mineralogical composition of the dusts was determined with an X ray powder diffraction (XRD) using a Panalytical Empyrean Powder Diffractometer equipped with a position sensitive X'Celerator detector. Cu K$\alpha$ radiation (k = 1.54178_A) at 40 kV and 30 mA. Scans were done over a 2h period, between 5° and 65° with an approximate step size of 0.033°.

**Acknowledgments**

We thank Dr. Yigal Erel and Ofir Tirosh from the Hebrew University of Jerusalem for their support in ICP-MS analyses, and Dr. Yael Kiro from Weismann Institute for conducting isotopic chromatography in her lab.

**References**

1.  I. Loladze, Hidden shift of the ionome of plants exposed to elevated CO2 depletes minerals at the base of human nutrition. *Elife* **2014** (2014).

2.  R. Lal, Soil degradation as a reason for inadequate human nutrition. *Food Secur* **1**, 45–57 (2009).

3.  H. Marschner, E. A. Kirkby, C. Engels, Importance of Cycling and Recycling of Mineral Nutrients within Plants for Growth and Development. *Botanica Acta* **110**, 265–273 (1997).

4.  D. S. Goll, *et al.*, Atmospheric phosphorus deposition amplifies carbon sinks in simulations of a tropical forest in Central Africa. *New Phytologist* **237**, 2054–2068 (2023).

5.  L. Van Langenhove, *et al.*, Atmospheric deposition of elements and its relevance for nutrient budgets of tropical forests. **149**, 175–193 (2020).

6.  G. S. Okin, N. Mahowald, O. A. Chadwick, P. Artaxo, Impact of desert dust on the biogeochemistry of phosphorus in terrestrial ecosystems. *Global Biogeochem Cycles* **18** (2004).

7.  A. Gross, *et al.*, Variability in Sources and Concentrations of Saharan Dust Phosphorus over the Atlantic Ocean. *Environ Sci Technol Lett* **2**, 31–37 (2015).

8.  O. A. Chadwick, L. A. Derry, P. M. Vitousek, B. J. Huebert, L. O. Hedin, Changing sources of nutrients during four million years of ecosystem development. *Nature* **397**, 491–497 (1999).

9.  D. Palchan, Y. Erel, M. Stein, Geochemical characterization of contemporary fine detritus in the Dead Sea watershed. *Chem Geol* **494**, 30–42 (2018).

10. N. K. Fageria, M. P. B. Filho, A. Moreira, C. M. Guimarães, Foliar Fertilization of Crop Plants. *http://dx.doi.org.mgs.ariel.ac.il/10.1080/01904160902872826* **32**, 1044–1064 (2009).

11. M. Ishfaq, *et al.*, Foliar nutrition: Potential and challenges under multifaceted agriculture. *Environ Exp Bot* **200** (2022).

12. L. J. Arvin, C. S. Riebe, S. M. Aciego, M. A. Blakowski, Global patterns of dust and bedrock nutrient supply to montane ecosystems. *Sci Adv* **3**, eaao1588 (2017).

13. A. Gross, S. Tiwari, I. Shtein, R. Erel, Direct foliar uptake of phosphorus from desert dust. *New Phytologist* **230**, 2213–2225 (2021).

14. A. Gojon, O. Cassan, L. Bach, L. Lejay, A. Martin, The decline of plant mineral nutrition under rising CO2: physiological and molecular aspects of a bad deal. *Trends Plant Sci* **28**, 185–198 (2023).

15. S. B. St.Clair, J. P. Lynch, The opening of Pandora's Box: climate change impacts on soil fertility and crop nutrition in developing countries. **335**, 101–115 (2010).

16. D. T. Clarkson, J. B. Hanson, THE MINERAL NUTRITION OF HIGHER PLANTS. *Ann. Rev. Plant Physiol* **31**, 239–98 (1980).

17. N. M. Lowe, The global challenge of hidden hunger: perspectives from the field. *Proceedings of the Nutrition Society* **80**, 283–289 (2021).

18. I. Loladze, Rising atmospheric CO2 and human nutrition: toward globally imbalanced plant stoichiometry? *Trends Ecol Evol* **17**, 457–461 (2002).

19. B. Langmann, Volcanic Ash versus Mineral Dust: Atmospheric Processing and Environmental and Climate Impacts. *ISRN Atmospheric Sciences* **2013**, 1–17 (2013).

20. J. F. Kok, *et al.*, Contribution of the world's main dust source regions to the global cycle of desert dust. *Atmos Chem Phys* **21**, 8169–8193 (2021).

21. T. T. N. Dam, *et al.*, X-ray Spectroscopic Quantification of Phosphorus Transformation in Saharan Dust during Trans-Atlantic Dust Transport. *Cite This: Environ. Sci. Technol* **55**, 12694–12703 (2021).

22. P. Hinsinger, Bioavailability of soil inorganic P in the rhizosphere as affected by root-induced chemical changes: A review. *Plant Soil* **237**, 173–195 (2001).

23. S. Muhammad, K. Wuyts, R. Samson, Atmospheric net particle accumulation on 96 plant species with contrasting morphological and anatomical leaf characteristics in a common garden experiment. *Atmos Environ* **202**, 328–344 (2019).

24. R. Van Oss, *et al.*, Genetic relationship in cicer Sp. expose evidence for geneflow between the cultigen and its wild progenitor. *PLoS One* **10** (2015).

25. H. B. Bradl, Adsorption of heavy metal ions on soils and soils constituents. *J Colloid Interface Sci* **277**, 1–18 (2004).

26. S. Tiwari, R. Erel, A. Gross, Chemical processes in receiving soils accelerate solubilisation of phosphorus from desert dust and fire ash. *Eur J Soil Sci* **73** (2022).

27. H. Lambers, *et al.*, "Nutrient-acquisition strategies" in *A Jewel in the Crown of a Global Biodiversity Hotspot.* , H. Lambers, Ed. (Kwongan Foundation and the Western Australian Naturalists' Club Inc., 2019).

28. S. Shakir, S. S. e. A. Zaidi, F. T. de Vries, S. Mansoor, Plant Genetic Networks Shaping Phyllosphere Microbial Community. *Trends in Genetics* **37**, 306–316 (2021).

29. M. Starr, T. Klein, A. Gross, Direct foliar acquisition of desert dust phosphorus fertilizes forest trees despite reducing photosynthesis. *Tree Physiol* **43**, 794–804 (2023).

30. M. Stein, S. L. Goldstein, From plume head to continental lithosphere in the Arabian-Nubian shield. *Nature* **382**, 773–778 (1996).

31. J. R. Farmer, *et al.*, Deep Atlantic Ocean carbon storage and the rise of 100,000-year glacial cycles. *Nature Geoscience 2019 12:5* **12**, 355–360 (2019).

32. S. M. Aciego, *et al.*, Dust outpaces bedrock in nutrient supply to montane forest ecosystems. *Nat Commun* **8**, 14800 (2017).

33. V. Masson-Delmotte, *et al.*, "Climate Change 2021: The Physical Science Basis Contribution of Working Group I to the Sixth Assessment Report of the Intergovernmental Panel on Climate Change" (2021) (May 30, 2023).

34. S. S. Myers, *et al.*, Increasing CO2 threatens human nutrition. *Nature 2014 510:7503* **510**, 139–142 (2014).

35. A. Gross, B. L. Turner, T. Goren, A. Berry, A. Angert, Tracing the Sources of Atmospheric Phosphorus Deposition to a Tropical Rain Forest in Panama Using Stable Oxygen Isotopes. *Environ Sci Technol* **50**, 1147–1156 (2016).

36. M. Bauters, *et al.*, Fire-derived phosphorus fertilization of African tropical forests. *Nat Commun* **12** (2021).

37. N. Mahowald, *et al.*, Global distribution of atmospheric phosphorus sources, concentrations and deposition rates, and anthropogenic impacts. *Global Biogeochem Cycles* **22** (2008).

38. R. Ciriminna, A. Scurria, G. Tizza, M. Pagliaro, Volcanic ash as multi-nutrient mineral fertilizer: Science and early applications. *JSFA Reports* **2**, 528–534 (2022).

39. A. Eger, P. C. Almond, L. M. Condron, Phosphorus fertilization by active dust deposition in a super-humid, temperate environment—Soil phosphorus fractionation and accession processes. *Global Biogeochem Cycles* **27**, 108–118 (2013).

40. C. Zhu, *et al.*, Carbon dioxide (CO2) levels this century will alter the protein, micronutrients, and vitamin content of rice grains with potential health consequences for the poorest rice-dependent countries. *Sci Adv* **4** (2018).

41. H. Sun, C. M. Weaver, Decreased Iron Intake Parallels Rising Iron Deficiency Anemia and Related Mortality Rates in the US Population. *J Nutr* **151**, 1947–1955 (2021).

42.    M. R. Smith, S. S. Myers, Impact of anthropogenic CO2 emissions on global human nutrition. *Nature Climate Change 2018 8:9* **8**, 834–839 (2018).

43.    R.S. Hilman, "Iron Deficiency and Other Hypoproliferative Anemias " in *Harrison's Principles of Internal Medicine*, 15th Ed., E. Braunwald, *et al.*, Eds. (McGraw-Hill Medical Publishing, 2001).

44.    S. R. Pasricha, J. Tye-Din, M. U. Muckenthaler, D. W. Swinkels, Iron deficiency. *The Lancet* **397**, 233–248 (2021).

45.    A. A. Caracioni, A. Wahl, S. Walsh, P. Silberstein, Iron Deficiency Anemia. *xPharm: The Comprehensive Pharmacology Reference*, 1–4 (2007).

46.    A. Mehri, Trace Elements in Human Nutrition (II) – An Update. *Int J Prev Med* **11** (2020).

47.    J. Sardans, A. Rivas-Ubach, J. Peñuelas, The C:N:P stoichiometry of organisms and ecosystems in a changing world: A review and perspectives. *Perspect Plant Ecol Evol Syst* **14**, 33–47 (2012).

48.    P. B. Reich, S. E. Hobbie, Decade-long soil nitrogen constraint on the CO2 fertilization of plant biomass. *Nature Climate Change 2012 3:3* **3**, 278–282 (2012).

49.    S. Zechmeister-Boltenstern, *et al.*, The application of ecological stoichiometry to plant–microbial–soil organic matter transformations. *Ecol Monogr* **85**, 133–155 (2015).

50.    O. Grau, *et al.*, Nutrient-cycling mechanisms other than the direct absorption from soil may control forest structure and dynamics in poor Amazonian soils. *Sci Rep* **7** (2017).

51.    R. Wang, *et al.*, Global forest carbon uptake due to nitrogen and phosphorus deposition from 1850 to 2100. *Glob Chang Biol* **23**, 4854–4872 (2017).

52. ,    Climate Change 2021 The Physical Science Basis. Contribution of Working Group I to the Sixth Assessment Report of the Intergovernmental Panel on Climate Change 2021.

53.    A. F. Longo, *et al.*, P-NEXFS analysis of aerosol phosphorus delivered to the Mediterranean Sea. *Geophys Res Lett* **41**, 4043–4049 (2014).

54.    Y. Nakamaru, M. Nanzyo, S. I. Yamasaki, Utilization of apatite in fresh volcanic ash by pigeonpea and chickpea. *Soil Sci Plant Nutr* **46**, 591–600 (2000).

55.    B. Langmann, Volcanic Ash versus Mineral Dust: Atmospheric Processing and Environmental and Climate Impacts. *ISRN Atmospheric Sciences* **2013**, 1–17 (2013).

56.    A. Stockdale, *et al.*, Understanding the nature of atmospheric acid processing of mineral dusts in supplying bioavailable phosphorus to the oceans. *Proceedings of the National Academy of Sciences* **113**, 14639–14644 (2016).

57. A. Gross, D. Palchan, M. D. Krom, A. Angert, Elemental and isotopic composition of surface soils from key Saharan dust sources. *Chem Geol* **442**, 54–61 (2016).

58. C. Guieu, *et al.*, Large clean mesocosms and simulated dust deposition: A new methodology to investigate responses of marine oligotrophic ecosystems to atmospheric inputs. *Biogeosciences* **7**, 2765–2784 (2010).

59. D. Palchan, M. Stein, A. Almogi-Labin, Y. Erel, S. L. Goldstein, Dust transport and synoptic conditions over the Sahara–Arabia deserts during the MIS6/5 and 2/1 transitions from grain-size, chemical and isotopic properties of Red Sea cores. *Earth Planet Sci Lett* **382**, 125–139 (2013).

60. T. Tanaka, *et al.*, JNdi-1: a neodymium isotopic reference in consistency with LaJolla neodymium. *Chem Geol* **168**, 279–281 (2000).

61. J. Jweda, L. Bolge, C. Class, S. L. Goldstein, High Precision Sr-Nd-Hf-Pb Isotopic Compositions of USGS Reference Material BCR-2. *Geostand Geoanal Res* **40**, 101–115 (2016).

62. G. J. Wasserburg, S. B. Jacobsen, D. J. DePaolo, M. T. McCulloch, T. Wen, Precise determination of Sm/Nd ratios, Sm and Nd isotopic abundances in standard solutions. *Geochim Cosmochim Acta* **45**, 2311–2323 (1981).

**Figures and Tables**

[Figure]

**Figure 1.** Biomass and P content increases due to dust application treatments at aCO$_2$ of 412ppm. (A) Image of experiment setting of the root treatment. (B) Image of experiment setting of foliar treatment. **(C)** Shoot biomass of root treated plants. **(D)** Shoot biomass of foliar treated plants. **(E)** Shoot P content of root treated plants. **(F)** Shoot P content of foliar treated plants. The asterisk denotes statistically significant difference from the control. The biomass and P content in the root treated plants do not show increases compared with the control groups. However, the foliar treatment of both desert dust and volcanic ash caused significant increases in the shoot biomass and P content. This implies that plants acquire P from fresh dust deposits on their foliage and not from the root system. Red color represents control plants, orange desert dust treatment and purple volcanic ash treatment.

[Figure]

**Figure 2.** Comparison of two chickpea varieties - CR934 (dotted, pink) and Zehavit (yellow) and their leaf properties under dust foliar fertilization. **(A)** Biomass and P uptake response to foliar dust P. Each column indicates the difference Δ (%) between the foliar dusted plants and the control untreated plants (n=6). **(B)** (I) Leaf surface pH. Each value indicates an average of five measurements on a plant throughout the growth season in control treatment (n=90), and two measurements in foliar dust treatment (n=10). One asterisk indicates significant differences between treatments using a T-test, and a one-way ANOVA (P≤0.05). Three asterisks indicate significant differences between treatments using a T-test, and a one-way ANOVA (P≤0.001). **(C)** Leaf Glandular (black column) and non-glandular (please add a dash to the Non glandular in the figure and change it to Non-glandular) (white column) trichrome density in CR934 and Zehavit control plants (-P and +P). Different letters indicate significant differences between varieties and treatments using Tukey-HSD test (P≤0.05) (n=12). Capital letters refer to non-glandular trichomes and small letters refer to glandular trichomes. **(D)**. SEM scans of non-glandular (red circles) and glandular (yellow circles) trichomes of typical Zehavit leaf. **(E)**. SEM scans of leaves of CR934 (left) and Zehavit (right) varieties. The Zehavit clearly shows higher density of trichomes in the abaxial surface, rendering it as more fit to extract nutrients from dust particles. **(F)**. Exudates of organic acids. Each column indicates the average of leaf washing from four plants, in -P control treatment (n=4). Two asterisks indicate significant differences between treatments using a T-test, and a one-way ANOVA (P≤0.01). Values are concentrations compared with an internal standard.

[Figure]

**Figure 3.** Quantification of dust mineral-nutrient flux from the foliage. Radiogenic isotopic ratios of $^{143}Nd/^{144}Nd$ in the different sample groups (x-axis) expressed in $\varepsilon Nd$ values. Diamonds represent the two applied mineral fractions of volcanic ash and desert dust; circles represent plants treated with the dusts and the control groups. Large circles represent plants growing in the 850 ppm eCO$_2$ and small circles represent the 412 ppm aCO$_2$. The color scale reflects the % contribution of Nd originating from the dusts via the foliage, which was calculated using a two-component mixing model. The control plants' Nd signature reflects the inheritance value from the seed, where a value of $\varepsilon Nd$=-0.3 is set as the control, $\varepsilon Nd$=-10.3 as the desert dust value, and $\varepsilon Nd$=4.6 as the volcanic ash value. A foliar contribution of more than 60% is evident in the plants applied with desert dust and more than 70% in the plants applied with volcanic ash. Standard errors on the isotopic values are all smaller than the depicted data points.

**Root treatment 850 ppm    Foliar treatment 850 ppm**

[Figure]

**Figure 4.** Biomass and P content increases due to dust application treatments at eCO$_2$ of 850ppm. **(A)** Shoot biomass of root treated plants. **(B)** shoot biomass of foliar treated plants. **(C)** Shoot P content of root treated plants. **(D)** Shoot P content of foliar treated plants. The asterisk denotes statistically significant difference from the control. The biomass and P content in the root treated plants do not show increases compared with the control groups. However, the foliar treatment of both desert dust and volcanic ash caused significant increases in the shoot biomass and P content. This implies that plants acquire P from fresh dust deposits on their foliage and not from the root system. Colors: red – control plants, orange – desert dust treatment, and purple – volcanic ash treatment.

[Figure]

[Figure]

**Figure 5.** Comparison of the % change in plant ionome of our experiments under various conditions compared with ambient CO2 control plants. Changes in nutrient concentrations of control $eCO_2$ plants (red circles) show that $eCO_2$ conditions deteriorate plant root uptake significantly. **(A)** The effect of foliar treatment of desert dust (orange triangles). **(B)** The effect of foliar treatment of volcanic ash (purple squares). Error bars denote SD. Both desert dust and volcanic ash treated plants show that the ability of plants to uptake nutrients from the foliage will replenish, and even increase, concentrations of Fe and Ni in a possible future earth, hence mitigating the plant nutrient reduction caused by elevated $CO_2$.

**Supplementary Text**

**Soil pot experiments**

We performed a parallel experiment under $aCO_2$ conditions to test whether our findings also apply for natural soil conditions. The conditions and experimental design were identical to the perlite medium experiments. The only difference was that instead of perlite we used local sandy "Hamra" soil that is common in the Israeli costal plains. Chemical analysis of the sandy soil showed that the total P concentration was very low (less than 50 μg per g), and mostly in unavailable Ca-P bounded forms. Six plants were fertigated with low P solution: two were used as the control group, two plants were applied with desert dust on their foliage, and two plants had desert dust applied to their roots. The soil was gently mixed to let the fresh dust settle deeper towards the roots system.

The results of this experiment confirmed the results of the main experiment. The root treated plants showed biomass and P content identical to the control group whereas the foliar treated plants show increases in biomass (**Fig. S1**). This experiment indicates that over a short timescale (few weeks), mineral nutrients from dust are not acquired via the roots system, rather they are only acquisitioned via the foliage.

**Dust holding capacity**

One of our challenges in conducting dust application experiments, is to quantify the dust interacting with the foliage, as not all the dust that was sprinkled remains on the leaves. Quantifying the dust that interacts with the foliage will allow to distinguish the plant variants that are more capable and fit to acquire mineral nutrients from their foliage. Furthermore, this quantification will enable to determine the optimal dust coverage/weight with minimum decreases in photosynthesis and maximum mineral nutrient acquisition.

The estimate that we used is the dust holding capacity, i.e., a visual evaluation of the dust coverage following application of 1g of dust. The values were given to plants by the same person and range between 0-5, where 0 had no dust visible on the leaves and 5 was fully covered by dust. This method can only be used within one species between different variants. In our chickpea lines experiment where we compared two variants, one wild type (CR934, inefficient) and a common domesticated variety ("Zehavit", efficient), we noticed differences between the two varieties where the CR934 received a value of 2 and Zehavit received a value of 3 (**Fig. S3**).

[Figure]

Fig. S1. Shoot biomass of experiment setting on sandy soil. Control plants (red, no dust application), plants that were applied with desert dust on the roots (yellow, root treatment), and on the foliage (orange, foliar treatment) that were grown on local natural soil. The foliar treated plants show an increase in biomass whereas the root treatment does not. This suggests that mineral nutrient uptake via the foliage is the only mechanism that enables plants to uptake nutrients from freshly deposited dust.

[Figure]

Fig. S2 Phyllosphere sugars profile when compared to an internal lab standard. Values on the y axis are relative to the standard.

[Figure]

Fig. S3. Dust holding capacity in two chickpea types. The wild CR934 showed less dust following dust application compared with the cultivated Zehavit. This is probably due to higher density of trichomes and their properties (see Fig. 2 in main paper).

---

## Author Response (AR2)

Dear editor and reviewers,

We are happy to resubmit our paper "Direct foliar phosphorus uptake from wildfire ash" (EGUSPHERE-2023-2617) after additional review of the paper.

Our responses to the reviewers are provided below in **bold.** For your convenience, following our responses, you will find the revised version of the manuscript with a "track changes" to make it easier for the reviewers to follow the changes we have made in the text.

**Editor comments**

Please change the y-axis label of Figure 4 to reflect the variable (and not just the unit; e.g. $CO_2$ effect on plant nutrient concentration (%)).

**R: Thank you for your suggestion. Corrected accordingly (P11 L314).**

**Report 1**

The authors have provided good responses to the reviewers' comments, and made appropriate changes to the manuscript. I'd just like to raise two points: Regarding the response to my comment on Figs 1 and 2 (page 19 of the authors' response), I believe that P concentration should be reported, at least in Supplementary Info. The inference "We did not detect changes in P concentration because any additional P was directed to biomass growth since the plants were P starved" should be explicitly stated in the text. Interested biologists will want to see at what P concentration this occurred, and non-biologists will probably not be familiar with the concept that additional P uptake does not always result in increased P concentration but can be invested in additional biomass with similar P concentration. My expectation would be that P concentration would increase AND growth would increase. It is hard to understand how photosynthesis would be boosted (Line 331) without an increase in P concentration, if indeed P did limit photosynthetic rate in these plants.

**R: We thank the reviewer for the suggestion. The P concentration of the plants is presented in tables S3 and S4 in the supplementary information. Additionally, we have included a mention of P starvation (P6, lines 231-232) and provided an explanation regarding biomass**

**gain, boosting of photosynthesis, and P concentration (P12, lines 332-337). The average concentration of P in the P starved plants ranged between 600-800 µg/g, and we assume that the additional P absorbed via the foliar pathway was directed towards biomass gain and photosynthesis rather than increasing the P concentration.**

In the new Fig. 1, both in the figure, its caption and in the main text, the units are wrong: 6000 mg P per g ash is impossible. You mean mg/kg or microg/g.

**R: Changed accordingly (P8 L260), (P8 L268).**

**Report 2**

Delete "that we" on line 226.

**R: Corrected accordingly (P6 L227).**

Change show to showed on line 230 to match the tenses.

**R: Corrected accordingly (P6 L233).**

It is difficult to see the median and IQR in Fig. 3. Please ensure that the final uploaded high-res version is clearer.

**R: We thank the reviewer for this comment. We widened the median and interquartile range lines and softened the pattern of the graphs. (P10 L338).**

The I in "In" should be capitalized on line 388.

R: **Corrected accordingly (P13 L396).**